# Cystine-knot peptide inhibitors of HTRA1 bind to a cryptic pocket within the active site region

Yanjie Li[1,4], Yuehua Wei[1,4], Mark Ultsch[2], Wei Li[1], Wanjian Tang[1], Benjamin Tombling[1], Xinxin Gao[1], Yoana Dimitrova[2], Christian Gampe[3], Jakob Fuhrmann[1], Yingnan Zhang[1], Rami N. Hannoush [1] ✉ & Daniel Kirchhofer [1] ✉

Cystine-knot peptides (CKPs) are naturally occurring peptides that exhibit exceptional chemical and proteolytic stability. We leveraged the CKP carboxypeptidase A1 inhibitor as a scaffold to construct phage-displayed CKP libraries and subsequently screened these collections against HTRA1, a trimeric serine protease implicated in age-related macular degeneration and osteoarthritis. The initial hits were optimized by using affinity maturation strategies to yield highly selective and potent picomolar inhibitors of HTRA1. Crystal structures, coupled with biochemical studies, reveal that the CKPs do not interact in a substrate-like manner but bind to a cryptic pocket at the S1' site region of HTRA1 and abolish catalysis by stabilizing a non-competent active site conformation. The opening and closing of this cryptic pocket is controlled by the gatekeeper residue V221, and its movement is facilitated by the absence of a constraining disulfide bond that is typically present in trypsin fold serine proteases, thereby explaining the remarkable selectivity of the CKPs. Our findings reveal an intriguing mechanism for modulating the activity of HTRA1, and highlight the utility of CKP-based phage display platforms in uncovering potent and selective inhibitors against challenging therapeutic targets.

Cystine-knot peptides (CKPs) are naturally occurring constrained polypeptide frameworks that have been shown to exhibit protease stability and resistance under various stresses[1–3]. They are typically characterized by three intertwined disulfide bonds that form a rigid knotted structure, displaying multiple surface loops that are available for binding to proteins. CKPs exhibit various biological activities towards multiple protein classes including proteases, ion channels, and various other receptors that are of pharmacological interest[4,5]. Recently, we employed a phage-displayed CKP library based on the *Ecballium elaterium* trypsin inhibitor II (EETI-II)[6]

scaffold to identify LRP6 receptor binders that potently inhibited Wnt signaling[7,8]. Here, we expanded the scaffold diversity by building a phage-displayed library based on the potato carboxypeptidase A1 inhibitor (CPI)[9,10]. We then applied both the EETI-II and CPI-based libraries for discovering inhibitors of the serine protease HTRA1 (High Temperature Requirement A1). We selected EETI-II and CPI as the scaffolds of choice because they are small in size with a molecular weight less than 3.5 kDa; yet, they exhibit high stability, solubility and can be produced either recombinantly or by peptide synthesis[3,7,11].

[1]Department of Early Discovery Biochemistry, Genentech Inc., 1 DNA Way, South San Francisco, CA 94080, USA. [2]Department of Structural Biology, Genentech Inc., 1 DNA Way, South San Francisco, CA 94080, USA. [3]Department of Discovery Chemistry, Genentech Inc., 1 DNA Way, South San Francisco, CA 94080, USA. [4]These authors contributed equally: Yanjie Li, Yuehua Wei. ✉e-mail: ramihannoush@gmail.com; dak@gene.com

HTRA1 is a trimeric serine protease and belongs to the small S1C subfamily of S1 family proteases (MEROPS peptidase database)[12], which have important roles in normal physiology[13–17]. HTRA1 regulates the TGF-β pathway during development and HTRA1 loss-of-function mutations lead to familial ischemic cerebral small-vessel disease[18]. HTRA1 enzyme activity was also linked to bone formation[19,20], the degradation of β-amyloid and tau in Alzheimer's disease[21–23] and to regulation of tumor progression[24–26], arthritis[27–32], osteoporosis[33] and the "wet" (neovascular) as well as "dry" (geographic atrophy) forms of age-related macular degeneration[34–36]. In addition, a chromosomal inversion leading to an in-frame SH3PXD2A-HTRA1 fusion protein was shown to increase the invasion and growth of Schwannoma, a common nerve sheath tumor[37]. Therefore, inhibition of HTRA1 activity could be of potential therapeutic benefit for several diseases, such as arthritis, osteoporosis, Schwannoma and age-related macular degeneration.

HTRA1 is composed of a central trypsin fold serine protease domain, which is flanked by the N-terminal N-domain (IGFBP-like and Kazal-like modules) and a C-terminal PDZ domain[38–40]. HTRA1 cleaves many substrates[41,42], including biglycan, decorin and Dickkopf-related protein 3 (DKK3)[43,44] and has no strict P1 residue specificity (Schechter and Berger[45] nomenclature for substrate P residues interacting with protease S sites) since it cleaves after various aliphatic and polar residues[38,40,44]. HTRA1 forms a homotrimer through its protease domains that positions the three active sites in close proximity such that antibodies are unable to simultaneously access them. Indeed, known inhibitory antibodies of HTRA1 bind to epitopes peripheral to the active site and use allosteric mechanisms[46,47]. In contrast, small compact peptides such as CKPs do not have these spatial restrictions, which increases the available surface area for functional binding, including the occupation of all three active sites within an HTRA1 trimer. These active sites are able to adopt different conformational states involving three specificity-determining loops, i.e., Loop1 [189-Loop] (chymotrypsin numbering in bracket), Loop2 [220-Loop] and LoopD [148-Loop][38,40,47]. The non-competent active site conformation can readily transition to the competent form and engage in catalysis[38]. This raises the possibility that CKPs may impair catalysis by trapping HTRA1 in the inactive conformation. Such a mechanism was recently described for the allosteric anti-HTRA1 Fab15H6.v4, which stabilized the non-competent active site conformation and completely inhibited catalysis[47]. This Fab was clinically tested for the treatment of geographic atrophy but, disappointingly, did not slow disease progression over a 72-week treatment period[48]. The other HTRA family members HTRA2-4, which also form homotrimers, share sequence and structural similarity with HTRA1[49,50], as well as conformational plasticity of the active site region[51–54], which poses a considerable challenge to achieve inhibitor selectivity.

In this work, we employed an expanded CKP diversity platform composed of two different CKP scaffolds to identify HTRA1 inhibitors. The potent CPI library-derived inhibitors lock the HTRA1 active site in a non-competent conformation and derive their exquisite selectivity by binding to a low homology site centered at the cryptic pocket of HTRA1. The most potent inhibitors of the CPI series engage in extended interactions with the two active site regions of two neighboring protomers in the HTRA1 trimer complex. These findings underscore the utility of CKP phage libraries with enriched scaffold diversity for the generation of selective inhibitors against multimeric complex proteins as exemplified by the serine protease HTRA1.

## Results

### Identification of HTRA1 inhibitors from phage display libraries using two different CKP scaffolds

We selected two CKPs, *Ecballium elaterium* trypsin inhibitor II (EETI-II)[6] and the potato carboxypeptidase A1 inhibitor (CPI)[9,10], as scaffolds for designing phage display libraries. We introduced amino acid randomization in two of the surface-exposed loops within the selected CKPs.

The EETI-II library with Loop1 and Loop5 diversity (Supplementary Fig. 1) was recently described[7]. The CPI library was constructed with sequence and length diversity in the neighboring Loops 2 and 5, which together may form a continuous binding region (Supplementary Fig. 1). These combined libraries, having sequence diversity of more than $10^{10}$, generated an extensive range of potential binding epitopes that increased the probability to discover inhibitors of HTRA1 enzyme function. Since the N- and the PDZ domains of HTRA1 do not contribute to enzyme activity[38,40], we used the HTRA1 protease domain trimer (HTRA1$^{PD(SA)}$; S328A mutant) for library screening. The identified binders were characterized by phage ELISA and the top hits were selected for peptide synthesis and further biochemical characterization (Supplementary Fig. 2a, b). We observed that none of the EETI-II derived peptides blocked full length HTRA1 (HTRA1$^{FL}$), although two of them showed weak inhibition of HTRA1$^{PD}$ as measured in HTRA1 enzyme assays (Supplementary Fig. 2c). On the other hand, the CPI-derived peptides 3B3 and 3A7 showed strong inhibition of both HTRA1 constructs. The amino acid composition of the Loop5 region of peptides 3A7 and 3B3 was different from that of the original CPI scaffold. Given that 3A7 and 3B3 shared a similar sequence in Loop5 (Supplementary Fig. 2b), we selected only one of them, 3B3, for further affinity maturation.

### Affinity maturation of 3B3 yields potent HTRA1 inhibitors

Because CPI interacts with the active site of carboxypeptidase A1 mainly through its Loop5 and the C-terminal region (PDB: 4CPA[10]), we reasoned that 3B3 may also use its C-terminal region to interact with HTRA1. Therefore, we generated two 3B3 libraries with fully randomized C-terminal regions having lengths of 5 (X5) and 8 (X8) amino acids (Fig. 1a). Panning of the pooled libraries against HTRA1$^{PD(SA)}$ yielded many improved binders by phage ELISA. The sequences of the hits showed a preference for bulky hydrophobic side chains in position 34 (X8 group) or the replacement of Y35 with glycine (X5 group) (Fig. 1a). For reasons that remain unclear, three peptides of the X8 group had an alanine to threonine change in the non-randomized loop4 at position 24 and this change was kept for subsequent synthesis. The synthesized peptides showed improved activity compared to the parent 3B3 (IC$_{50}$ = 53 nM) in enzyme assays using the synthetic substrate H2-OPT (Fig. 1a). The five most potent inhibitors had IC$_{50}$ values in the range of 1.6–7.6 nM, an improvement of 7–33-fold over the parent peptide. The inhibition curves of a peptide subset, representing the X5 and X8 groups, are shown in comparison to the parent 3B3 and the CPI scaffold (Fig. 1b). These five CKPs also strongly inhibited endogenous full-length HTRA1 secreted by human melanoma cells[46] (Supplementary Fig. 3a). The binding affinities measured by surface plasmon resonance (SPR) ($K_D$ range 0.7–7.0 nM and $K_D$ of 34 nM for the parent 3B3) (Fig. 1c and Supplementary Table 1) were in good agreement with the inhibitory potencies. The two most potent inhibitors 1A8 and 2D5 had very slow off-rates, probably due to favorable interactions of their C-terminal regions with HTRA1. It is noteworthy that the CPI scaffold neither inhibited HTRA1$^{FL}$ activity (Fig. 1b), nor bound to HTRA1$^{PD(SA)}$ in SPR experiments (Fig. 1c). Conversely, both 3B3 and its "sibling" 3A7 inhibited the CPI target enzyme carboxypeptidase A1 (Fig. 1d), most likely because they retained the C-terminal CPI sequence, which is important for carboxypeptidase A1 binding[10]. Consistent with this interpretation, the four affinity-matured CKPs with changed C-terminal sequences no longer inhibited carboxypeptidase A1 (Fig. 1d).

HTRA1 cleaves numerous protein substrates, including the ocular biomarker DKK3[44]. We chose DKK3, biglycan and decorin[43] as macromolecular HTRA1 substrates to assess the CKPs' inhibitory activities. Under the experimental conditions in which HTRA1$^{FL}$ completely degraded DKK3, biglycan and decorin, the five examined CKPs were able to fully inhibit the in vitro cleavage of these substrates, while the CPI scaffold was not inhibitory (Fig. 2a). Identical results were obtained

## a

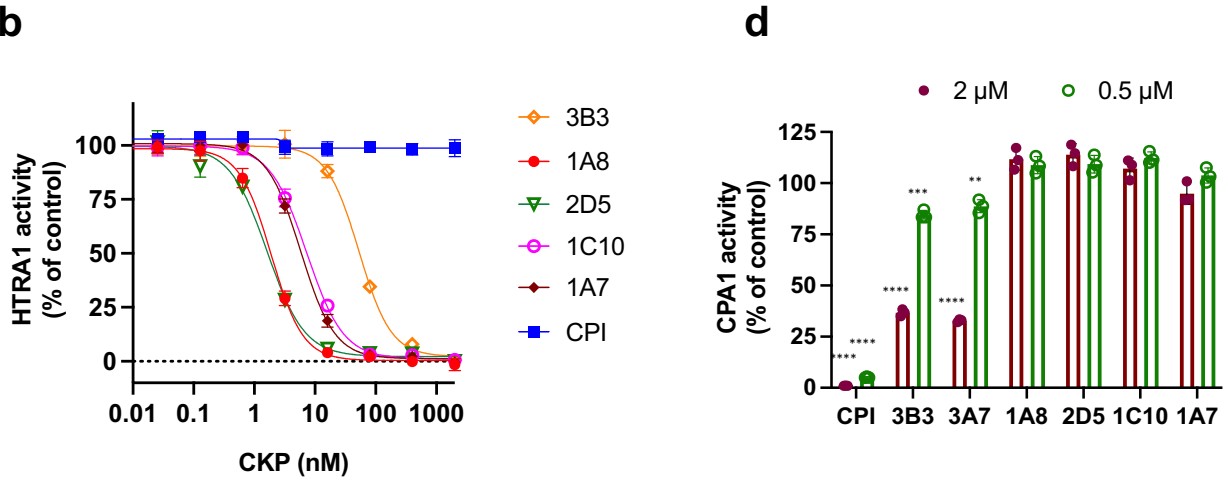

| Peptide | IC50 ± S.D. (nM) | 1 | 2 | 3 | 4 | 5 | 6 | 7 | 8 | 9 | 10 | 11 | 12 | 13 | 14 | 15 | 16 | 17 | 18 | 19 | 20 | 21 | 22 | 23 | 24 | 25 | 26 | 27 | 28 | 29 | 30 | 31 | 32 | 33 | 34 | 35 | 36 | 37 | 38 | 39 | 40 |
|---|---|---|---|---|---|---|---|---|---|---|---|---|---|---|---|---|---|---|---|---|---|---|---|---|---|---|---|---|---|---|---|---|---|---|---|---|---|---|---|---|---|
| | | N terminus | | | | | | L1 | | | | L2 | | | | | | L3 | | | | | | L4 | | | L5 | | | | | | | C terminus | | | | | | | |
| 3B3 (parent) | 53.1 ± 1.5 | H | A | D | P | I | C | N | K | P | C | K | T | H | D | D | C | S | G | A | W | F | C | Q | A | C | Y | Y | A | T | W | S | C | G | P | Y | V | G | - | - | - |
| X5 library | | H | A | D | P | I | C | N | K | P | C | K | T | H | D | D | C | S | G | A | W | F | C | Q | A | C | Y | Y | A | T | W | S | C | X | X | X | X | X | - | - | - |
| X8 library | | H | A | D | P | I | C | N | K | P | C | K | T | H | D | D | C | S | G | A | W | F | C | Q | A | C | Y | Y | A | T | W | S | C | X | X | X | X | X | X | X | X |
| *X8 group (X8 library):* | | | | | | | | | | | | | | | | | | | | | | | | | | | | | | | | | | | | | | | | | |
| 1C12 | 7.6 ± 0.1 | H | A | D | P | I | C | N | K | P | C | K | T | H | D | D | C | S | G | A | W | F | C | Q | A | C | Y | Y | A | T | W | S | C | G | Y | Y | P | Y | W | F | R |
| 1E4 | 27.8 ± 0.7 | H | A | D | P | I | C | N | K | P | C | K | T | H | D | D | C | S | G | A | W | F | C | Q | T | C | Y | Y | A | T | W | S | C | G | L | G | Y | R | N | A | S |
| 1A8 | 1.9 ± 0.2 | H | A | D | P | I | C | N | K | P | C | K | T | H | D | D | C | S | G | A | W | F | C | Q | T | C | Y | Y | A | T | W | S | C | G | W | G | L | R | Q | I | D |
| 2D5 | 1.6 ± 0.1 | H | A | D | P | I | C | N | K | P | C | K | T | H | D | D | C | S | G | A | W | F | C | Q | T | C | Y | Y | A | T | W | S | C | Q | W | P | P | R | H | R | D |
| *X5 group (X5 library):* | | | | | | | | | | | | | | | | | | | | | | | | | | | | | | | | | | | | | | | | | |
| 2C10 | 17.5 ± 1.4 | H | A | D | P | I | C | N | K | P | C | K | T | H | D | D | C | S | G | A | W | F | C | Q | A | C | Y | Y | A | T | W | S | C | G | P | G | Q | P | - | - | - |
| 2G1 | 24.2 ± 1.6 | H | A | D | P | I | C | N | K | P | C | K | T | H | D | D | C | S | G | A | W | F | C | Q | A | C | Y | Y | A | T | W | S | C | G | P | G | A | G | - | - | - |
| 1B12 | 22.0 ± 0.7 | H | A | D | P | I | C | N | K | P | C | K | T | H | D | D | C | S | G | A | W | F | C | Q | A | C | Y | Y | A | T | W | S | C | G | P | G | Q | I | - | - | - |
| 1H8 | 27.9 ± 1.5 | H | A | D | P | I | C | N | K | P | C | K | T | H | D | D | C | S | G | A | W | F | C | Q | A | C | Y | Y | A | T | W | S | C | G | P | G | R | V | - | - | - |
| 2F11 | 12.6 ± 0.6 | H | A | D | P | I | C | N | K | P | C | K | T | H | D | D | C | S | G | A | W | F | C | Q | A | C | Y | Y | A | T | W | S | C | G | P | G | Y | K | - | - | - |
| 1C10 | 7.3 ± 0.4 | H | A | D | P | I | C | N | K | P | C | K | T | H | D | D | C | S | G | A | W | F | C | Q | A | C | Y | Y | A | T | W | S | C | G | P | G | R | Y | - | - | - |
| 1A7 | 5.8 ± 0.3 | H | A | D | P | I | C | N | K | P | C | K | T | H | D | D | C | S | G | A | W | F | C | Q | A | C | Y | Y | A | T | W | S | C | G | P | G | W | R | - | - | - |

## b

## d

## c

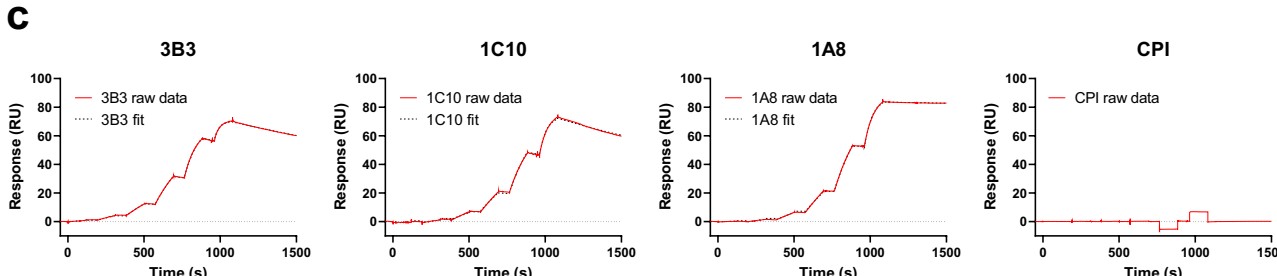

**Fig. 1 | Inhibitory activities and binding kinetics of CKPs obtained from affinity maturation of 3B3. a** Sequences and IC$_{50}$ values in HTRA1$^{FL}$ enzyme inhibition assays of the parent 3B3 and of affinity-matured CKPs (X5 and X8 group) derived from the X5 and X8 libraries with five and eight randomized residues at the C terminus (indicated as "X"); L1–L5, Loop1–Loop5. **b** CKPs of both the X8 and X5 groups fully inhibit HTRA1$^{FL}$ activity, while the CPI (carboxypeptidase A1 inhibitor) scaffold has no activity. **c** SPR sensorgrams showing kinetics of HTRA1$^{PD(SA)}$ binding of parent 3B3, the CPI scaffold, 1C10 (X5 group) and 1A8 (X8 group). **d** CPI, 3B3, and

3A7 have the same C terminus and inhibit carboxypeptidase A1 (CPA1), but the affinity-matured CKPs (X5 and X8 group) with their modified C-termini no longer inhibit CPA1. ** $P = 0.0037$; *** $P = 0.0002$; **** $P < 0.0001$; two-sided unpaired Student's $t$ test comparing CKPs with DMSO control. Values in (**a**) and the graphs (**b** and **d**) show the mean ± S.D. of $n = 3$ independent experiments except $n = 10$ for 1A8 in (**a**) and (**b**). Sensorgrams in (**c**) are representative of three independent experiments.

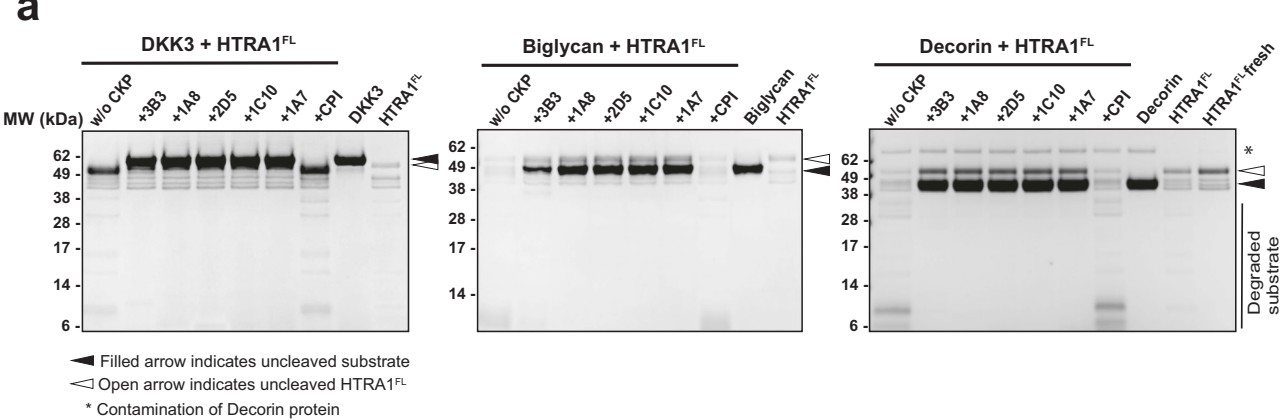

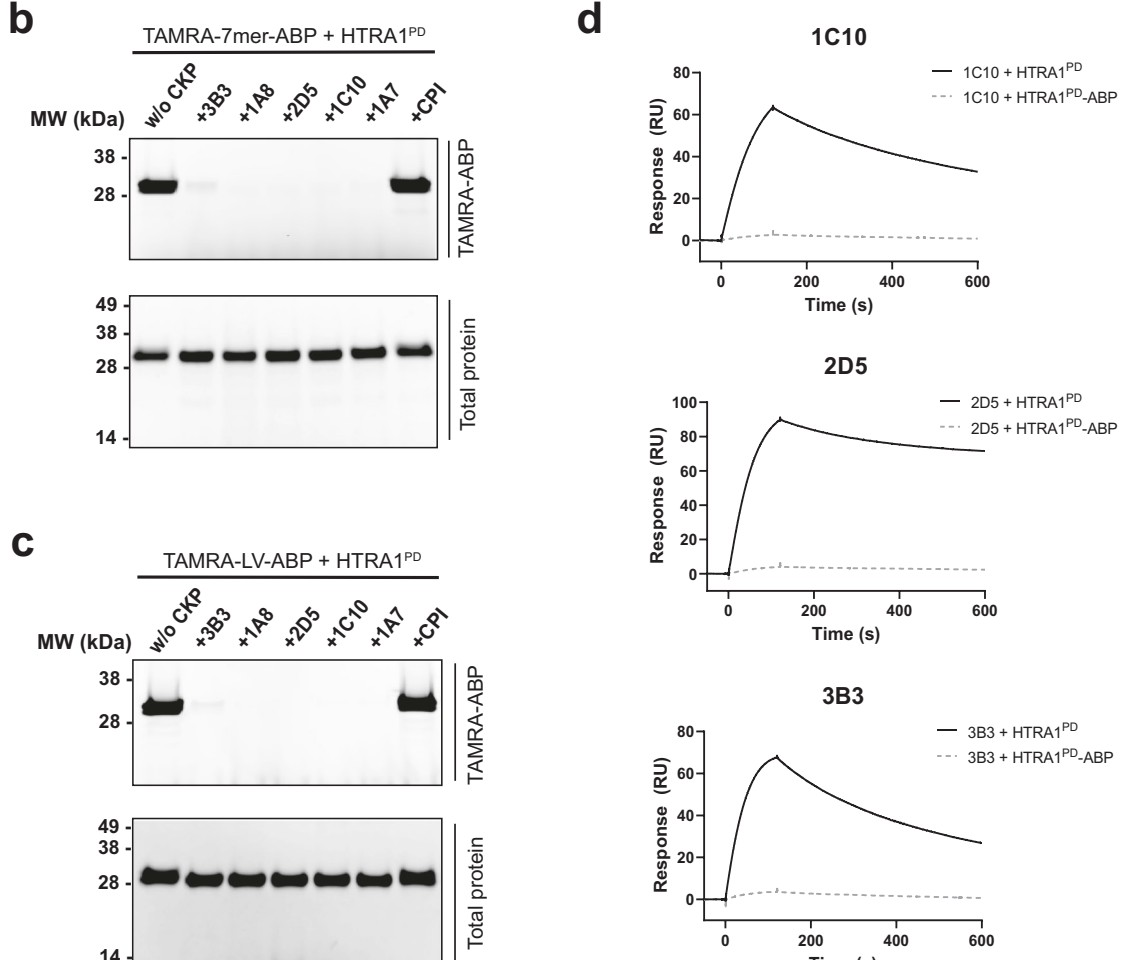

**Fig. 2 | Inhibition of macromolecular substrate cleavage and inhibitory mechanism of affinity-matured CKPs. a** The parent 3B3 and affinity-matured CKPs of the X5 and X8 group completely inhibit HTRA1$^{FL}$-mediated cleavage of the macromolecular substrates DKK3, biglycan and decorin, while the CPI scaffold has no effect. During incubation in the absence of CKPs the HTRA1$^{FL}$ underwent degradation, as shown by the degradation of the main band (open arrows; 50 kDa) in comparison to intact HTRA1$^{FL}$ loaded onto the gel without prior incubation ("HTRA1$^{FL}$ fresh", right lane in "decorin" panel). This is likely due to auto-cleavage, since HTRA1$^{FL}$ inhibition by CKPs prevented degradation, which is best seen in the right panel ("decorin"). Labeling of HTRA1$^{PD}$ by the two fluorescent activity-based probes TAMRA-7mer-ABP (**b**) and TAMRA-LV-ABP (**c**). The CKPs completely prevented HTRA1$^{PD}$ labeling by the larger (**b**) and smaller (**c**) TAMRA probes (upper panels in **b** and **c**), while the CPI scaffold was not inhibitory. Lower panels are Coomassie-stained gels showing HTRA1$^{PD}$ loading for each lane. **d** Sensorgram showing that the CKPs 1C10, 2D5 and 3B3 no longer bound to HTRA1$^{PD}$ whose active site was occupied by the covalently bound 7mer-ABP (HTRA1$^{PD}$-ABP) (dotted lines). HTRA1$^{PD}$-ABP as well as untreated HTRA1$^{PD}$ were captured via their His-tags on CM5 sensor chips and the CKPs were tested at 3 μM (3B3) or 300 nM (1C10, 2D5). Images in (**a**–**c**) and binding kinetics in (**d**) are representative of at least three independent experiments.

with the N-terminal domain-deleted HTRA1 (HTRA1$^{PD/PDZ}$) (Supplementary Fig. 3b), which corresponds to an autocatalytically produced HTRA1 species found in-vivo[55,56]. Collectively, the results suggested that the HTRA1 inhibitory activity of the CKPs is independent of the type of substrate or HTRA1 species.

## Biochemical studies to investigate the inhibitory mechanism
To elucidate the mechanism of inhibition of the CKPs, we carried out biochemical and enzymatic studies. First, we determined whether the CKPs impede HTRA1 active site labeling by two fluorescent activity-based probes (ABP) with phosphonate warheads, the longer TAMRA-DPMFKLV-ABP (TAMRA-7mer-ABP)[47] and the shorter TAMRA-LV-ABP[44]. These probes occupy the S1 pocket and different portions of the S subsites region (Schechter and Berger[45] nomenclature for substrate binding subsites). The TAMRA-7mer-ABP strongly labeled HTRA1$^{PD}$, but each of the five tested CKPs including the parent 3B3 abrogated labeling, whereas the CPI scaffold had no effect (Fig. 2b). Identical results were obtained with the smaller activity-based probe TAMRA-LV-ABP (Fig. 2c). Based on these results, we hypothesized that the CKPs could bind to the HTRA1 active site at, or near the S1 pocket. Therefore, we used SPR to measure the binding of CKP to HTRA1$^{PD}$ in which the active site was occupied with the covalently attached 7mer-ABP. The examined CKPs 1C10 and 2D5 from the X5 and X8 groups, respectively, as well as the parent 3B3 were no longer able to bind to the active site-blocked HTRA1$^{PD}$ (Fig. 2d). These studies suggested that the CKPs interfered with fundamental aspects of HTRA1 catalysis, most likely by binding to the active site region of the intact HTRA1 trimer.

## Crystal structure reveals that the CKPs bind to a cryptic binding pocket of HTRA1 and stabilize a non-competent active site
To understand the molecular basis of the inhibitory mechanism, we first determined the X-ray crystal structure of the parent 3B3 in complex with HTRA1$^{PD(SA)}$ at 3.1 Å resolution (PDB: 8SDM, Supplementary Table 2). In agreement with the biochemical studies, the structure showed that each of the three active sites in the HTRA1$^{PD(SA)}$ trimer was occupied by one CKP (Fig. 3a) forming a 3:1 complex (3B3:HTRA1 trimer). The three copies of 3B3 in this complex, designated chains I, X, and Y, lacked electron density (Supplementary Fig. 4a) for the N-terminal residues H1-D3 (chain X and I), H1-A2 (chain Y) and for the C-terminal residue G37 (chain I). All three chains maintained the disulfide bond network and the fold of the parent CPI scaffold (PDB: 4CPA) (vs chain I: r.m.s.d. of 0.59 Å for 28 atoms) and showed no noticeable deviations in loop structures or side chain rotamers (Supplementary Fig. 5a). For further discussion, we chose the lowest B factor 3B3 molecule, chain I. The main interface of 3B3 with the HTRA1 protomer chain A is located in the region of the S′ subsites of HTRA1 (Fig. 3a). Residues Y26, Y27 and W30 of Loop5 account for half (365 Å$^2$) of the total buried 3B3 surface area of 694 Å$^2$ (Fig. 3b). Remarkably, Loop2 which was hard-randomized in the initial CPI library design, but remained unchanged in both 3B3 and 3A7 binders, is also contacting HTRA1 through T12 and H13. The C terminus of 3B3 is far away from the active site (Fig. 3b), contrasting with the parent CPI in which the C terminus inserts into the active site of carboxypeptidase A1. Therefore, our rationale for generating C-terminal extension libraries that was based on the carboxypeptidase A1-CPI paradigm, turned out to be false, but fortuitously led to potent inhibitors.

The key binding residue Y27 of the 3B3-Loop5 inserts into a hydrophobic pocket situated between the catalytic H220 and the β-strand stem region of HTRA1-LoopA [37-Loop] (Fig. 3b). This HTRA1 hydrophobic pocket is in close proximity to the S1′ subsite, as indicated by the superimposed co-crystal structure of the trypsin fold serine protease thrombin with bound PAR-1 peptide substrate[57] (Supplementary Fig. 5b, c). The inserted Y27 is further stabilized by W30 that binds close to the rim of the Y27 pocket (Fig. 3b, c). This pocket has not been observed in other HTRA1 structures exemplified by PDB:

3TJN[38] (Fig. 3c–f), and therefore is cryptic in nature and able to transition from the closed to the open form to accommodate Y27 of 3B3. The conformational state of the pocket is governed by the gatekeeper residue V221, which fills the pocket in the closed state (Fig. 3e, f). The pocket opening is accompanied by a remarkable movement of V221 (5.3 Å shift for V221$^{Cγ}$ in all protein chains and 2.5 Å for V221$^{Cα}$), which relocates to the approximate position that is occupied by the V222 residue in the closed conformation (Fig. 3d, f). There are additional but less dramatic movements by residues of the ascending (L188) and descending (A202, S203, G204) β-strands connecting to LoopA [37-Loop]. From the cryptic pocket binding site, the 3B3-Loop5 extends to the center of the active site and stabilizes a non-competent active site conformation (Fig. 3g and Supplementary Fig. 5d). The same non-competent conformation has been observed in previously published apo-HTRA1 crystal structures (PDB: 3NUM-chain A, 3TJO-chain X, NWU and 3TJN-chain A) and deviates from the competent state in several aspects. The active site HTRA1 Loops 1 and 2 [189- and 220-Loop] are conformationally different with Loop2-residue L345 occluding the S1 pocket, the oxyanion hole is non-functional and the flipped side chain of the catalytic H220 has moved away from the catalytic S328 (A328 in 3B3 structure), precluding its function as hydrogen acceptor for catalysis (Supplementary Fig. 5d). The conformation of the non-functional oxyanion hole (G326-N327-S328A) (Supplementary Fig. 5d) is stabilized by the 3B3 residues Y26 and P34, which pack against HTRA1-Y325 (Fig. 3g). In addition, the flipped side chain of L345 is held in place by the two 3B3 residues A28 and T29, and A28 is sandwiched between the catalytic H220 and S328 to obliterate catalysis. The loop ensemble that defines the non-competent state also includes LoopD [148-Loop], which is in an extended conformation compared to the competent state (Supplementary Fig. 5e). The same non-competent active site conformation was observed in a recent CryoEM structure of HTRA1$^{PD(SA)}$ in complex with the allosteric Fab15H6.v4 that binds to the distant HTRA1-LoopA [37-Loop] (Fig. 3h)[47]. Thus, the two HTRA1 inhibitors 3B3 and Fab15H6.v4 both stabilize the same non-competent state, but use fundamentally different mechanisms, which are orthosteric and allosteric in nature, respectively.

## Features of the cryptic binding pocket explain the excellent selectivity of CKPs towards HTRA1
The protease domains of the other human HTRA family members HTRA2-4 display high sequence conservation with HTRA1 (similarity of 70%, 81%, and 76% for HTRA2, HTRA3, and HTRA4, respectively), as well as structural similarity[49,50] (AlphaFold model of HTRA4[58,59]). To assess the potential cross-reactivity of the CKPs with other members of the HTRA family, we performed enzymatic and SPR binding assays. In assays with the synthetic HTRA1-4 substrate H2-OPT[38,49,60,61] the panel of five CKPs inhibited HTRA1$^{PD/PDZ}$ but not HTRA2-4$^{PD/PDZ}$ (Fig. 4a). As expected, the CPI scaffold had no inhibitory effect. Identical results were obtained with the full-length forms of HTRA1,3,4, confirming the excellent selectivity of the CKPs (Supplementary Fig. 6a). The same experiments were carried out with the macromolecular substrate casein (casein-BODIPY® assay), which is cleaved by all HTRA proteases. The results matched those with the H2-OPT substrate in that the CKPs neither affected proteolytic activity of HTRA2-4$^{PD/PDZ}$, nor of the full-length forms HTRA3-4$^{FL}$ (Supplementary Fig. 6b, c). To further confirm inhibitor selectivity, we carried out SPR binding assays, which demonstrated that neither the parent 3B3, nor 1C10 or 2D5 representing the X5 and X8 groups, showed any detectable binding to HTRA2-4$^{PD/PDZ}$ (Fig. 4b and Supplementary Fig. 7).

The complex structure of 3B3 with HTRA1 allowed us to explore the reasons for the observed CKP selectivity within the HTRA family. Sequence alignment of HTRA1-4 showed significant residue differences between HTRA1 and HTRA2-4 at key 3B3 contact sites in the region of the Y27 pocket (Supplementary Fig. 8a). In addition, HTRA3 has a long LoopB insertion, which is likely detrimental to CKP binding.

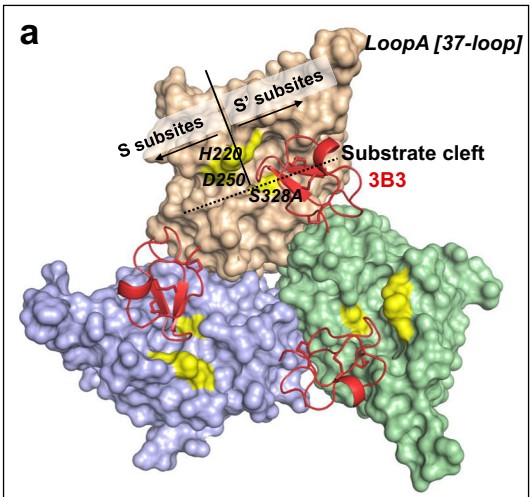

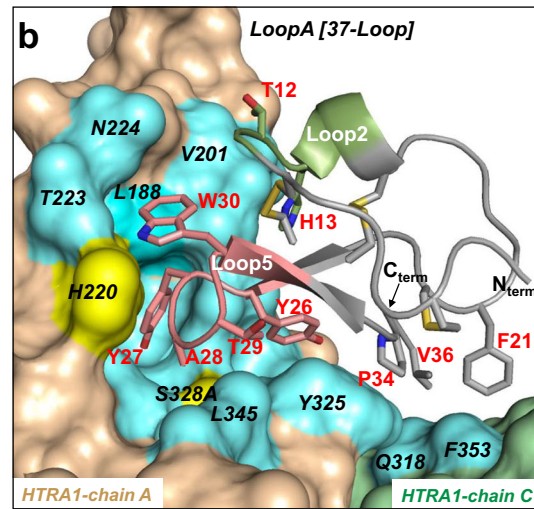

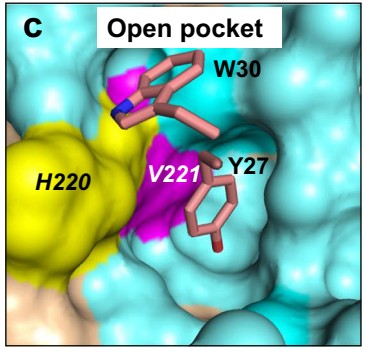

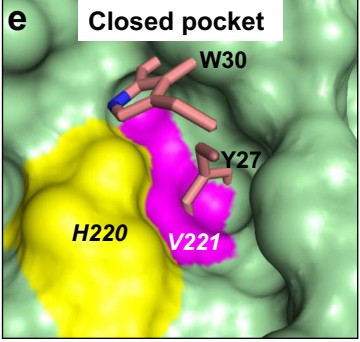

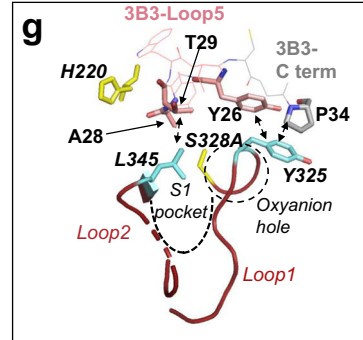

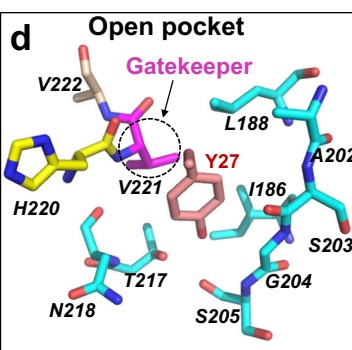

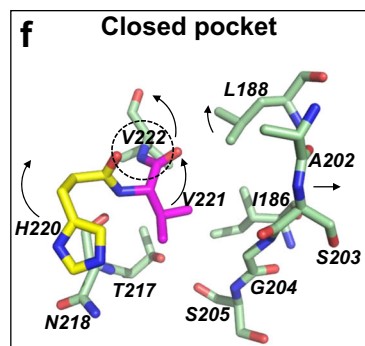

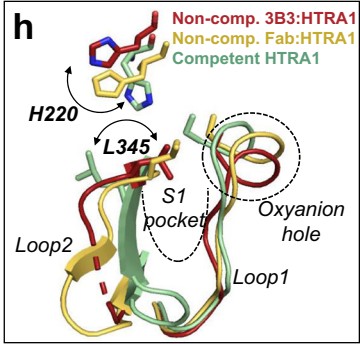

**Fig. 3 | Crystal structure of the 3B3:HTRA1$^{PD(SA)}$ complex shows the cryptic pocket and non-competent active site. a** Overview of 3B3:HTRA1$^{PD(SA)}$ complex (HTRA1$^{PD(SA)}$, protease domain with active site S328A mutation). Surface representation of the HTRA1 protomers (brown, green and blue) with the three 3B3 peptides (red, cartoon) bound to the S' region between the catalytic triad (D250, H220, S328A in yellow) and LoopA. The approximate path of the substrate binding cleft and the S and S' subsite regions are indicated. **b** Close-up of the binding interface between 3B3 (chain I) with HTRA1 protomer (chain A, brown) and minor contacts with HTRA1-chain C. Contacts within 4 Å of 3B3 on HTRA1 (surface representation) are shown in cyan and are labeled in black italics (catalytic H220 and S328A in yellow). The contact residues on 3B3 (within 4 Å from HTRA1) are labelled in red and the two main binding loops are in salmon (Loop5) and green (Loop2). 3B3 also makes a minor contact to the neighboring HTRA1 protomer (chain C). **c** and **d** The open Y27-pocket of the 3B3:HTRA1$^{PD(SA)}$ complex (3B3 contact residues in cyan as in (**b**), gatekeeper V221 in magenta, catalytic H220 in yellow) as surface representation (**c**) or with key pocket residues as sticks (**d**); the 3B3 residues Y27 and W30 are shown as salmon sticks. **e** and **f** The closed Y27-pocket of apo-HTRA1 (PDB: 3TJN-chain B[38]) with HTRA1 in surface representation (**e**) or with key

pocket residues as sticks (**f**) (HTRA1 in green). The superimposed 3B3 residues Y27 and W30 (as salmon sticks) clash with the closed pocket (**e**), which is filled by V221 (magenta). In the closed-to-open transition (**f** and **d**) the gatekeeper residue V221 (magenta) moves away to open the pocket and allowing Y27 insertion. In the open pocket the gatekeeper V221 is in almost the same location as V222 in the closed conformation (dotted circles). Movements of the residues, including those of the β-strand (A202-S205), are indicated by arrows. **g** Interactions of 3B3 residues (sticks) that stabilize the non-competent active site conformation of HTRA1. The 3B3 residues Y26 and P34 contact HTRA1-Y325 (cyan) which stabilizes a distorted conformation of the oxyanion hole (dotted circle) and 3B3 residues A28 and T29 stabilize the flipped conformation of L345 (cyan), which occludes the S1 specificity pocket. The HTRA1-Loop2 residues T348 and A349 are not resolved in the structure and are shown as dotted line. **h** Comparison of the non-competent active site conformations of the 3B3:HTRA1 complex (red) and the Fab15H6.v4:H-TRA1 complex (PDB: 7SJN-chain A; yellow) with the competent conformation of apo-HTRA1 (PDB: 3TJN-chain B; green). The H220 and L345 residues are shown as sticks and the S1 pocket and oxyanion hole are indicated as dotted lines.

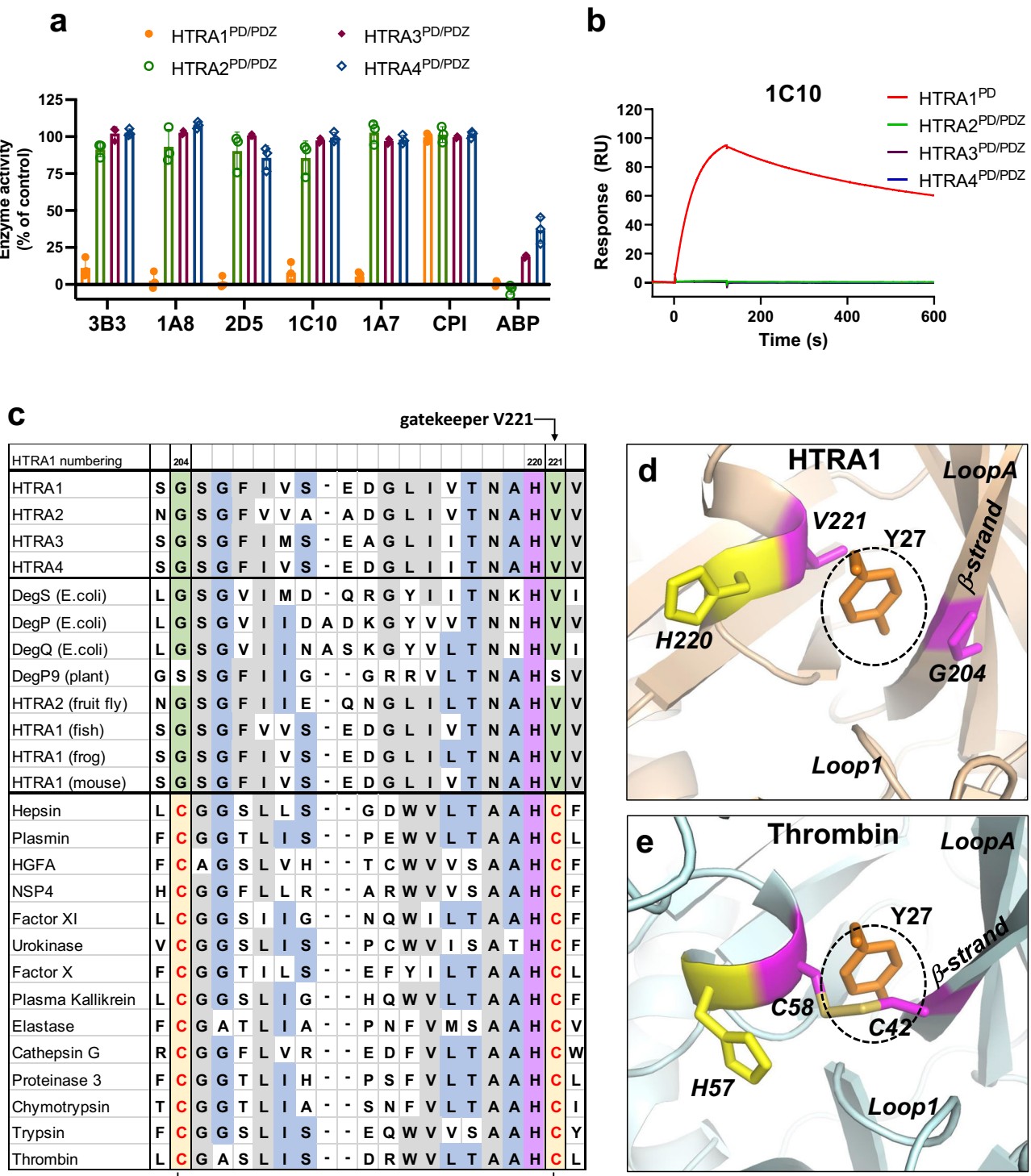

In contrast, murine HTRA1 (mHTRA1) has fully conserved contact residues and is inhibited by 3B3 (Supplementary Fig. 8b). To confirm the importance of these residues in affording 3B3 selectivity we chose HTRA4, which had the fewest changes at the 3B3 binding site and engineered a mutated HTRA4 protease that might become susceptible to 3B3 inhibition. The bulky HTRA4-Y200 residue, which replaces HTRA1-A202 is a conspicuous impediment for 3B3 binding. Therefore, we performed an alanine-tyrosine swap and generated the chimeric proteins HTRA4$^{PD/PDZ}$-Y200A and HTRA1$^{PD}$-A202Y (Supplementary Fig. 8c). While HTRA1$^{PD}$ enzymatic activity was reduced with increasing

3B3 concentrations, the HTRA1$^{PD}$-A202Y chimera became completely resistant to inhibition (Supplementary Fig. 8d), likely caused by a steric clash of Y202 with 3B3-Loop5. Conversely, HTRA4$^{PD/PDZ}$ was not inhibited by 3B3, but the HTRA4$^{PD/PDZ}$-Y200A chimera became susceptible to 3B3 inhibition reaching a 57% reduction of enzyme activity (Supplementary Fig. 8d). These findings support the view that the cryptic pocket site provides CKP selectivity.

Trypsin fold serine proteases of the S1A subfamily, such as trypsin and thrombin, have three canonical disulfide bonds ([C42-C58], [C168-C182], [C191-C220])[62]. These disulfides are missing in HTRA1 as well as

**Fig. 4 | The cryptic binding pocket provides selectivity of CKPs for HTRA1. a** Selectivity of CKPs within the HTRA family. At 2 μM concentration the parent 3B3 and affinity-matured CKPs only inhibit HTRA1$^{PD/PDZ}$ but not HTRA2-4$^{PD/PDZ}$ in the H2-OPT enzyme assay. The 7mer activity-based probe (ABP) was used as a positive control inhibitor. Values are the mean ± S.D. of at least three independent experiments. There was no statistically significant reduction in the activities of HTRA2-4$^{PD/PDZ}$ by the CKPs vs DMSO controls (two-sided, unpaired Student's *t* test). HTRA1$^{PD/PDZ}$ was significantly inhibited by CKPs (*P* < 0.0001) except by the CPI scaffold. All HTRA proteases were significantly inhibited by the ABP (P = 0.0003 for HTRA4$^{PD/PDZ}$ and P < 0.0001 for HTRA1-3$^{PD/PDZ}$). **b** SPR sensorgrams showing binding kinetics of 1C10 (1 μM) to HTRA family members (data are representative of three independent experiments). **c** Sequence alignment of trypsin fold serine proteases of subfamilies S1C (HTRA proteases) and of S1A, clan PA (Hepsin, plasmin etc.). HTRA proteases DegS, DegP and DegQ of *Escherichia coli* (strain K12), DegP9 of *Arabidopsis thaliana* (plant), HTRA2 of *Drosophila*

*melanogaster* (fruit fly), HTRA1 of *Oryzias latipes* (Japanese rice fish), HTRA1 of *Xenopus laevis* (African clawed frog) and HTRA1 of *Mus musculus* (mouse). The canonical cysteine residues [C42-C58] of S1A proteases (yellow) are replaced with glycine and valine in HTRA proteases (green). Conserved residues within the two subfamilies are highlighted in gray and those conserved across both subfamilies in blue; the catalytic histidine is in purple. The gatekeeper residue V221 of HTRA1 is indicated by an arrow. Sequence alignment was carried out by using UniProt[78]. **d** The region of the cryptic pocket (dotted circle) in the 3B3:HTRA1$^{PD(SA)}$ complex (HTRA1 as brown cartoon) with 3B3 pocket residue Y27 in orange sticks. The gatekeeper residue V221 (magenta; in "open" position) and G204 (magenta sticks) of the β-strand descending from LoopA correspond to the canonical disulfide-forming residues [C58] and [C42], respectively (catalytic H220 is in yellow). **e** The canonical [C42-C58] disulfide bridge of thrombin (light blue cartoon) (PDB: 3LU9 with bound PAR-1 peptide removed) takes up the space where the cryptic pocket of HTRA1 is located (indicated by dotted circle and superimposed Y27 of 3B3).

in other HTRA family members, which also have a trypsin fold. Of particular relevance is the disulfide bond [C42-C58] with C58 positioned next to the catalytic histidine. A sequence alignment of trypsin fold serine proteases shows that in HTRA1, C58 is replaced with the gatekeeper V221 and C42 with G204 and, with few exceptions (e.g., DegP9 from plant), both the valine and glycine residues are conserved in HTRA proteases from bacteria to mammals (Fig. 4c). In the S1A subfamily, here exemplified by thrombin, this disulfide bond is precisely at the location of the cryptic pocket. We can conclude that none of the disulfide bond-containing S1A proteases can form this pocket, which precludes any interaction with the important CKP residue Y27 (Fig. 4d, e). In agreement, 3B3 and 1A8 did not inhibit the enzymatic activities of a panel of 14 trypsin-fold serine proteases of the S1A subfamily (Supplementary Fig. 8e).

### The structure of the 3A7: HTRA1$^{PD(SA)}$ complex indicates conformational plasticity of the cryptic binding pocket

3A7 was identified in the initial library screen (Supplementary Fig. 2b and c) and binds to HTRA1 with low nanomolar affinity (Supplementary Table 1). It differs from 3B3 only by three residues in Loop5, the most important of them, F27 replacing the key binding residue Y27 of 3B3. The structure of the 3A7:HTRA1$^{PD(SA)}$ complex was determined to 2.9 Å resolution (PDB: 8SDP, Supplementary Table 2) and is almost identical to that of the 3B3: HTRA1$^{PD(SA)}$ complex, forming the same crystal packing (r.m.s.d. of 0.36 Å for 592 atoms of the entire trimer complex) (Supplementary Fig. 9a). The reference chain 3A7-chain X superimposes well with the two other 3A7 copies of the complex, as well as with 3B3 (Supplementary Figs. 4b, 9b and 9c). The binding interface between 3A7 and HTRA1$^{PD(SA)}$ was similar to that of the 3B3:HTRA1$^{PD(SA)}$ complex (Supplementary Fig. 9d), but differed in one important feature. The cryptic pocket is now occupied by F27, which is positioned closer to the β-strands connecting to LoopA, allowing the N218 side chain to shift towards the pocket and contact the aromatic ring of F27 (Fig. 5a). This movement changes the shape and the size of the pocket. The gatekeeper residue V221 is in the same position as in the 3B3 structure (Fig. 5a), suggesting that the conformational changes during pocket opening are identical. The F27 residue is the main HTRA1 contact residue and together with Y26 and W30 constitute 50% of the total buried 3A7 surface area of 741 Å². The other two amino acid changes in Loop5, N29 and R31, either functionally replace the corresponding 3B3 residue (N29) or do not engage in any HTRA1 interactions (R31). The entire active site region adopts the same non-competent conformation as in the 3B3:HTRA1$^{PD(SA)}$ structure (Fig. 5b) with the now fully resolved Loop2 perfectly superimposing with the non-competent apo-form of HTRA1 (PDB: 3TJN-chain A)[38]. In conclusion, the different shapes of the cryptic pocket when occupied by tyrosine or phenylalanine indicated structural plasticity of the pocket, raising the prospect of improving this interaction using distinct inhibitor designs to increase affinity.

### Structure-based designs and affinity maturation

Guided by the structures we designed variants to improve interactions of the two key binding CKP residues Y26 and Y/F27 by using unnatural amino acids, many of which had halogen atoms introduced to the aromatic rings to enhance van der Waals interactions. In addition, K11 in Loop2 was substituted by charged amino acids to form a potential salt bridge interaction with HTRA1-K225 as well as R190 and, based on Ramachandran plot analysis, the Loop2 residue D14 was α-methylated to increase stability. The changes were incorporated into 1A8, which only differs from 3B3 in its C-terminal sequence. The results of HTRA1 enzymatic assays showed that two of the 12 synthesized variants had a slightly improved potency in enzymatic assays, i.e., the charge-reversed mutants 1A8-K11E and 1A8-K11D (Supplementary Fig. 10a). Some of the designs were well tolerated, but many of those intended for better interactions with the cryptic pocket (residue 27) had potency losses of greater than 10-fold. This initial SAR analysis indicated that the cryptic Y27 pocket only accepts small modifications such as 4-fluoro-phenylalanine and seems to have a size limit that restricts larger aromatic substituents such as 4-chloro-phenylalanine or the expanded homophenylalanine variant from efficient binding.

Furthermore, based on the structural information, we constructed libraries of randomized 3B3-Loop1, 2, and 5. The improved binders were derived from libraries of Loop1 and 5 and the top four sequences were incorporated into 1A8 for peptide synthesis and tested in enzymatic assays. However, only the CKP with the Loop5-T29N mutation showed a small improvement (2-fold) in potency (Supplementary Fig. 10a). The increased activity could be due to stabilizing intramolecular H-bonds between the N29 side chain and S31 (Supplementary Fig. 10b).

### Phage display strategies based on the crystal structure of the 1A8:HTRA1 complex lead to CKPs with improved potency

The X-ray crystal structure of the potent inhibitor 1A8 in complex with HTRA1$^{PD(SA)}$ was solved to 3.0 Å resolution (PDB: 8SE7, Supplementary Table 2). The crystal packing was vastly different than the previously solved complexes with 1A8 having four trimeric complexes arranged in a cage-like structure in the asymmetric unit. The three bound 1A8 chains superimposed well with each other and with the parent 3B3 (Supplementary Fig. 11a and b). Akin to the 3B3 and 3A7 complexes, 1A8 engaged in similar HTRA1 interactions, including the cryptic pocket (Fig. 5c) and stabilized the same non-competent active site conformation (Supplementary Fig. 11c). However, 1A8 made additional contacts to the neighboring HTRA1 protomer (chain A) through its C-terminal residues W34 and F21 (Fig. 5c). W34 functionally substituted for P34 of 3B3 and 3A7 with respect to the interaction with HTRA1-Y325, but it additionally contacted residues of the adjoining HTRA1 protomer together with the F21 residue of 1A8-Loop2 (Fig. 5d). These secondary contacts amount to 22% (201 Å²) of the total buried surface area (960 Å²), which is now substantially larger than that of 3B3 and

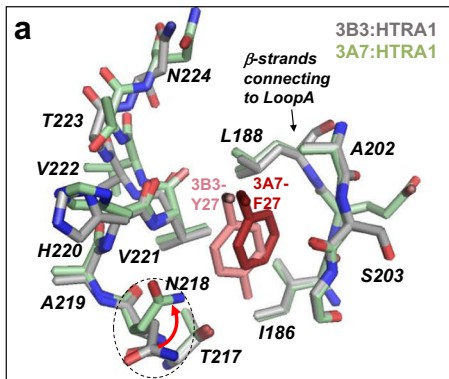

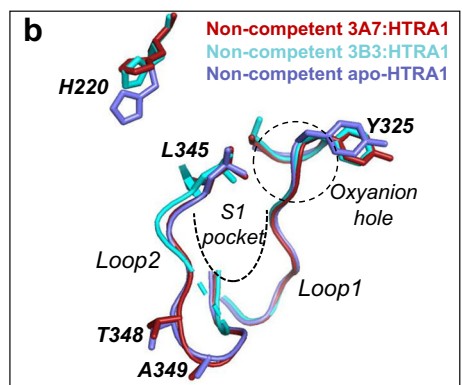

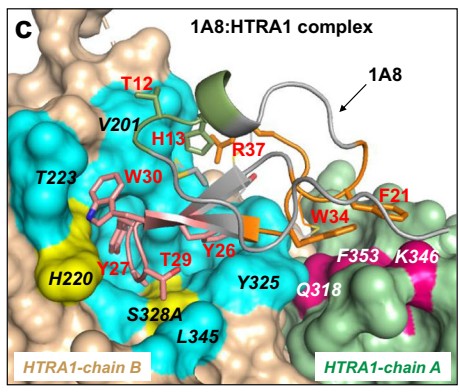

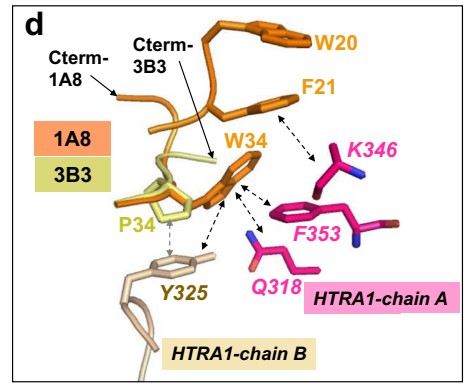

**Fig. 5 | The 3A7:HTRA1$^{PD(SA)}$ and 1A8: HTRA1$^{PD/(SA)}$ complexes. a** The cryptic HTRA1 pocket of the 3A7:HTRA1$^{PD(SA)}$ complex (green sticks) in comparison to the 3B3:HTRA1$^{PD(SA)}$ complex (gray sticks). The 3A7 residue F27, which replaces Y27 of 3B3, is closer to the LoopA-connecting β-strands, allowing the repositioning of the N218 side chain (red arrow, dotted circle), which changes the shape of the pocket. **b** Cartoon representation of the non-competent HTRA1 active site of 3A7:HTRA1$^{PD(SA)}$ (red) and of 3B3: HTRA1$^{PD(SA)}$ (cyan) in comparison to the non-competent active site of apo-HTRA1 (PDB: 3TJN-chain A) (blue). The entire Loop2 of the 3A7:HTRA1$^{PD(SA)}$ complex is resolved including the residues T348 and A349, which were missing in the 3B3 structure, and Loop2 superimposes perfectly with the non-competent conformation of the apo-HTRA1 reference structure. **c** The 1A8: HTRA1$^{PD(SA)}$ complex. Shown are the main interactions of 1A8 (chain X) with HTRA1

(chain B; brown) and additional contacts (W34 and F21; orange sticks) with the neighboring HTRA1-chain A (green). The residues of the two HTRA1$^{PD(SA)}$ protomers (surface representation) within 4 Å distance of 1A8 are shown in cyan and magenta, and the main binding loops of 1A8 (cartoon representation) are in salmon (Loop5) and green (Loop2). 1A8 also interacts with the catalytic triad residues H220 and S328A (yellow). Contact residues of 1A8 are labeled in red and those of HTRA1 are in black and in white italics. **d** The W34 of 1A8 (orange) makes important contacts to Y325 of HTRA1-chain B (brown) and to Q318 and F353 (magenta) of the neighboring protomer HTRA1-chain A. An additional contact to the neighboring protomer comes from F21, which is sandwiched between W34 and W20 (orange), and which contacts K346 of HTRA1-chain A. HTRA1 residues in Fig. 5 are in italics.

explains the improved binding affinity of 1A8. The 1A8 structure further deviates from 3B3 in that its longer C terminus turns in the opposite direction (Fig. 5d) and bends toward HTRA1 allowing the R37 residue to form electrostatic interactions with HTRA1 (Supplementary Fig. 11d). The 1A8 N-terminal region ($^1$HADPI$^5$) of chain X is fully resolved (except for the histidine side chain which is disordered) (Supplementary Fig. 4c and 11a,b) and projects towards the active site region of the neighboring HTRA1 protomer whose active site Loop2 is within 5 Å of the H1 residue (Fig. 6a). This raised the possibility of improving the binding affinity by constructing 1A8-based phage libraires with an extended N-terminal region that could reach the neighboring HTRA1 active site (Fig. 6a).

Phage displaying 1A8 on coat protein p3 bound tightly to HTRA1, making detection of affinity-improved clones technically challenging. Therefore, we synthesized 1A8 peptides with mutations of the Y27 and W30 residues to reduce binding affinity. The 1A8-Y27L mutant had a moderate activity loss in HTRA1 enzyme assays with an IC$_{50}$ value comparable to 3B3 (Fig. 6b) and was selected for phage library construction. Moreover, N-terminal deletion experiments indicated that the first two residues of 1A8 were not essential for activity (Fig. 6c) and, therefore, the N-terminal extension libraries included randomization of these first two residues. Based on the 1A8 structure, we also generated C-terminal libraries extending from the important W34 residue.

Both libraries yielded improved HTRA1 binders in phage ELISA assays and the top 12 sequences were synthesized as CKPs in which the L27 residue was changed back to the wildtype Y27 (Fig. 6d). Many CKPs showed improved potency in HTRA1$^{FL}$ enzyme assays with IC$_{50}$ values in the picomolar range (Fig. 6d). The N-terminal library-derived 1G10 was the most potent inhibitor (IC$_{50}$ 0.66 nM) and its extended N terminus started with a tyrosine (−1 position) that is also present in many other potent hits from this library (Fig. 6d). The top hit 2A6 from the C-terminal library had a short 5-residue C terminus in which the L36 of the parent 1A8 was replaced by I36 (Fig. 6d). SPR experiments demonstrated that 1G10 and 2A6 did not show any detectable binding to HTRA2-4 at concentrations >1000-fold above the determined IC$_{50}$ values (Supplementary Fig. 12a), confirming that the N- and C-terminal changes did not affect selectivity.

To gain insight into HTRA1 interactions by the extended N-terminal regions we solved the X-ray crystal structure of the 1G10:HTRA1$^{PD(SA)}$ complex (PDB: 8SE8, Supplementary Table 2). The crystal packing of the complexes was the same as seen for the parent 1A8 with four trimers arranged in a cage-like structure (Supplementary Fig. 12b). Unfortunately, none of bound 1G10 peptides had a fully resolved N terminus (Supplementary Fig. 4d), and there were no direct contacts observed with the neighboring active site were identified. However, two 1G10 peptides (chain X and chain U) were only missing

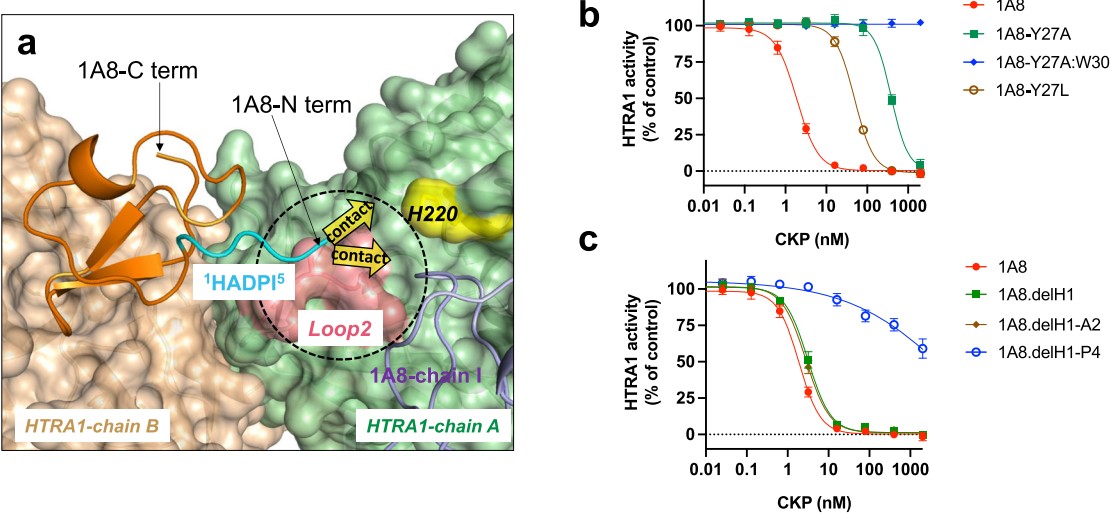

**d**

| Peptide | IC50 ± S.D. (nM) | -6 | -5 | -4 | -3 | -2 | -1 | 1 | 2 | 3 | 4 | 5 | 6 | _ | 32 | 33 | 34 | 35 | 36 | 37 | 38 | 39 | 40 |
|---|---|---|---|---|---|---|---|---|---|---|---|---|---|---|---|---|---|---|---|---|---|---|---|
|  |  | \<-- N terminus --\> |  |  |  |  |  |  |  |  |  |  |  |  | \<-- C terminus --\> |  |  |  |  |  |  |  |  |
| 1A8 | 1.90 ± 0.15 |  |  |  |  |  |  | H | A | D | P | I | C | _ | C | G | W | G | L | R | Q | I | D |
| *N-term. extensions* |  |  |  |  |  |  |  |  |  |  |  |  |  |  |  |  |  |  |  |  |  |  |  |
| 1G10 | 0.66 ± 0.12 |  |  |  |  |  | Y | P | V | D | P | I | C | _ | C | G | W | G | L | R | Q | I | D |
| 1H10 | 0.74 ± 0.14 |  |  |  |  |  | Y | R | R | D | P | I | C | _ | C | G | W | G | L | R | Q | I | D |
| 1D1 | 1.30 ± 0.18 |  |  |  |  |  | Y | S | V | D | P | I | C | _ | C | G | W | G | L | R | Q | I | D |
| 1B4 | 2.02 ± 0.31 |  |  |  |  |  | Y | T | P | D | P | I | C | _ | C | G | W | G | L | R | Q | I | D |
| 1G7 | 0.79 ± 0.09 |  |  |  |  |  | Y | S | P | D | P | I | C | _ | C | G | W | G | L | R | Q | I | D |
| 1G11 | 0.98 ± 0.01 |  |  |  |  |  | Y | P | R | D | P | I | C | _ | C | G | W | G | L | R | Q | I | D |
| 1D3 | 0.87 ± 0.23 |  |  |  |  |  | I | W | P | D | P | I | C | _ | C | G | W | G | L | R | Q | I | D |
| 1H8 | 2.16 ± 0.33 |  |  | S | I | S | W | Y | N | D | P | I | C | _ | C | G | W | G | L | R | Q | I | D |
| 1B12 | 9.73 ± 1.57 |  | A | R | S | N | Q | S | L | D | P | I | C | _ | C | G | W | G | L | R | Q | I | D |
| 1B5 | 1.05 ± 0.07 | R | H | F | S | Y | Y | P | F | D | P | I | C | _ | C | G | W | G | L | R | Q | I | D |
| *C-term. extensions* |  |  |  |  |  |  |  |  |  |  |  |  |  |  |  |  |  |  |  |  |  |  |  |
| 2G12 | 1.16 ± 0.09 |  |  |  |  |  |  | H | A | D | P | I | C | _ | C | G | W | P | P | R | E | L | G |
| 2A6 | 0.95 ± 0.17 |  |  |  |  |  |  | H | A | D | P | I | C | _ | C | G | W | G | I | R |  |  |  |

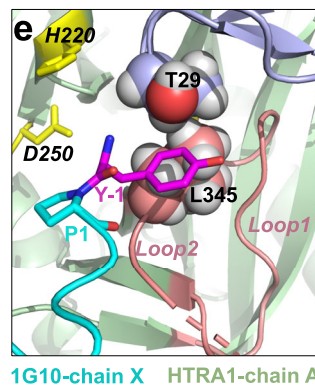

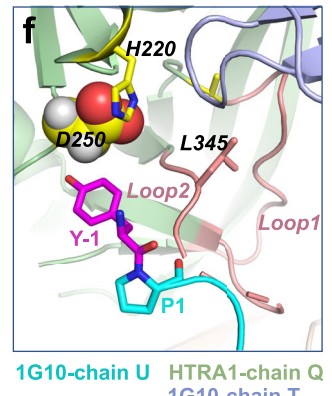

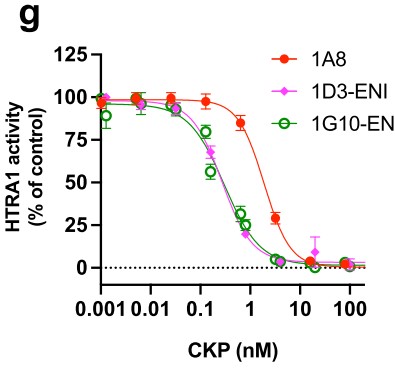

the N-terminal tyrosine residue and showed the second residue, proline, in a favorable position for modeling the N-terminal tyrosine. In 1G10-chain X the tyrosine side chain in the lowest energy conformation engages in hydrophobic interactions with the S1 pocket-occluding L345 situated in the active site Loop2 of the neighboring HTRA1 protomer (chain A) and with T29 of 1G10 (chain I) bound to the neighboring HTRA1 (Fig. 6e). In 1G10-chain U the tyrosine side chain only makes a minor hydrophobic contact by extending towards the catalytic D250 (Fig. 6f). Despite the apparent inherent flexibility of the N terminus, these two N-terminal conformers represent plausible, transient binding interactions with the neighboring active site.

Lastly, we synthesized a panel of 1G10 and 1D3-based CKPs (Supplementary Fig. 12c), which incorporated several improved amino acid changes, including K11E, T29N (Supplementary Fig. 10a) and L36I (Fig. 6d). The majority of the combinations did not show improvements in HTRA1$^{FL}$ enzyme assays (Supplementary Fig. 12c). However,

**Fig. 6 | Improving 1A8 potency by extending its N terminus towards the active site of the neighboring HTRA1 protomer. a** 1A8 (orange, cartoon) mainly interacts with HTRA1-chain B (brown, surface representation), but its fully resolved N terminus (¹HADPI⁵, cyan) projects towards Loop2 (salmon) of the active site of the neighboring HTRA1 protomer (chain A, green, surface representation). Extending the N terminus (arrows) may create new contacts to the neighboring active site centered at the catalytic H220 (yellow). Also shown is the 1A8-chain I (light purple) interacting with the active site of HTRA1-chain A. **b** and **c** HTRA1$^{FL}$ enzyme assays with 1A8 variants having Y27/W30 mutations to weaken binding affinity (**b**) or having N-terminal deletions (**c**). The determined IC$_{50}$ values ± S.D. were: 1A8 1.9 ± 0.2 nM, Y27A 389 ± 31 nM, Y27A:W30A > 2000 nM, Y27L 49.9 ± 3.8 nM, delH1 3.1 ± 0.5 nM, delH1-A2 2.7 ± 0.4 nM, delH1-P4 > 2000 nM. **d** HTRA1$^{FL}$ inhibition (IC$_{50}$) and sequences of synthesized peptides identified from screening phage libraries displaying 1A8-Y27L with N-terminal and C-terminal extensions. All peptides contained the wildtype Y27 residue. In some N-terminal extension peptides a cysteine residue was changed to a serine (bold underlined). Tyrosine was the preferred residue at the -1 position (boxed) of the N-terminal sequences. Amino acid changes compared to the parent 1A8 are in red. **e** Model based on the structure of the 1G10:HTRA1$^{PD(SA)}$ complex showing the N-terminal tyrosine residue (sticks, magenta) of the N-terminal region of 1G10-chain X (cyan) interacting with L345 (side chain as spheres) of the active site (Loop1 and 2 in salmon) of the neighboring HTRA1 protomer (chain A, light green). It also makes hydrophobic contacts with T29 (side chain as spheres) of the neighboring 1G10 peptide (chain I; light purple). Catalytic residues are shown as yellow sticks. **f** Model of the N-terminal tyrosine residue (sticks, magenta) of the N-terminal region of 1G10-chain U (cyan) interacting with the catalytic D250 (spheres) of the neighboring HTRA1 protomer (chain Q). Colors are as in (**e**). **g** Improved inhibition of HTRA1$^{FL}$ by 1D3- and 1G10-derived CKPs combining optimal residue changes (see Supplementary Fig. 12c for sequences). IC$_{50}$ values for 1A8, 1D3-ENI and 1G10-ENI were 1.90 ± 0.15 nM, 0.27 ± 0.02 nM and 0.30 ± 0.05 nM, respectively. Values in (**b, c, d** and **g**) are the mean ± S.D. of at least three independent experiments (n = 10 for 1A8 in (**b, c, d** and **g**), n = 5 for 1G10-ENI in (**g**), n = 4 for 1G10, 1H10, 1G7, 1G11, 1D3, and 2A6 in (**d**), and *n* = 3 for all other CKPs).

1D3-ENI and 1G10-ENI, which combined all changes (Supplementary Fig. 12c), had superior potencies and inhibited HTRA1$^{FL}$ in a concentration-dependent manner with IC$_{50}$ values of 0.27 nM and 0.30 nM, respectively (Fig. 6g), which were consistent with the determined $K_D$ values of 0.38 and 0.44 nM, respectively (Supplementary Table 1).

## Discussion

Phage-displayed libraries based on two natural CKP scaffolds were screened against the challenging therapeutic target HTRA1 which features three conserved active sites that are situated in close proximity. The CPI library-derived initial hits underwent several rounds of affinity maturation that were guided by structural information, yielding HTRA1 enzyme inhibitors with excellent potency and selectivity. The X-ray crystal structures revealed an orthosteric inhibition mechanism in which the CKPs compete with substrate binding and stabilize a non-competent active site conformation[38,40]. The conformational constraints imposed on the active site loops by the CKPs disabled the loops from transitioning back to the enzymatically competent state, resulting in the complete abolishment of catalysis, as evidenced by HTRA1 enzyme assays with various substrates and activity-based probes.

The inhibition mechanism of the CKPs is fundamentally different from that employed by the majority of naturally occurring serine protease inhibitors, which bind to their cognate protease(s) in a substrate-like manner by inserting the reactive site loop into the catalytic center[63,64], which is described as the "standard mechanism"[65]. The CKPs neither bind in a substrate-like manner nor have a reactive site loop. Instead, they bind to a cryptic pocket at the S1' site region of HTRA1 to disrupt catalysis and substrate interaction with the catalytic center. The cryptic pocket was a key feature of the inhibition mechanism by providing binding affinity and, most importantly, by conferring excellent selectivity. Inhibitor selectivity for trypsin fold serine proteases is challenging to achieve due the high sequence and structural homology within the protease families. The active site conservation of HTRA proteases is exemplified by our observation that the employed fluorescence-quenched substrate H2-OPT[60] is efficiently cleaved by HTRA1-3 and that the 7mer-ABP was not selective for HTRA1 but also inhibited the other HTRA proteases (Fig. 4a), similar to the non-selective inhibition of HTRA proteases by peptidic active site inhibitors[66]. The reason for the selectivity of the CKP inhibitors is that the main binding site was not located at the highly conserved catalytic center, but at the less conserved S' region centered at the cryptic pocket. Several CKP interaction residues are not preserved in HTRA2-4, exemplified by the HTRA4 residue Y200, which upon incorporation into HTRA1 (HTRA1$^{PD}$-A202Y), conferred resistance to 3B3 inhibition.

Beyond the HTRA protease family, the CKP inhibitor selectivity extends to the large S1A subfamily of trypsin fold serine proteases owing to the canonical disulfide bond ([C42-C58]) located at the site of the cryptic pocket in HTRA1. The presence of this disulfide bond in the canonical trypsin fold serine proteases (S1A subfamily) precludes CKP binding, in accordance with experimental data of a selected protease panel, indicating exceptional selectivity of the CKP inhibitors. On the other hand, the absence of this constraining disulfide bond in HTRA1 may afford increased conformational plasticity, allowing an unimpeded movement of the gatekeeper V221 residue to open the cryptic pocket. Remarkably, this gatekeeper residue is evolutionarily conserved in HTRA proteases from bacteria to humans. Because all HTRA proteases are missing the canonical disulfide bonds, this raises the possibility that the cryptic pocket may be a general property of HTRA proteases. We admit the highly speculative nature of this proposition, but we find some support in the observations that 3B3 inhibited mouse HTRA1 and the mutant HTRA4$^{PD/PDZ}$-Y200A. Since the cryptic pocket is the main binding site for 3B3, we interpret these results to indicate that both mouse HTRA1 and the HTRA4 mutant were able to form the pocket.

While this cryptic pocket was not observed in published structures of HTRA1, HTRA2 and HTRA3[38,40,49,50], there was a brief description in the supplementary section by Eigenbrot et al. [38] of an unusual protomer in one of the apo-HTRA1 trimers showing a β-octylglucoside, a mild detergent from the crystallization solution, inserted into a deep pocket. Although the depth and shape of this pocket differs from that described herein, both pocket entrances are at the same location. However, despite the apparent plasticity of this pocket, our attempts to obtain highly potent inhibitors using unnatural amino acid substitutions to improve pocket interaction were not successful. The breakthrough came with the 1A8 structure, showing the 5-residue N terminus in close proximity to the adjoining HTRA1 protomer, which inspired an N-terminal extension phage display strategy. Many of the obtained inhibitors, including the CKP 1G10, had picomolar potencies and showed a strong preference for a tyrosine at the 6th N-terminal residue (−1 position). According to models based on the 1G10:HTRA1$^{PD(SA)}$ structure, the tyrosine contacts the active site of the neighboring HTRA1 protomer, rendering 1G10 a bi-partite inhibitor that interacts with two distinct active sites within the HTRA1 trimer. The 1G10- and 1D3-derived CKPs, 1G10-ENI and 1D3-ENI, were the two most potent inhibitors (IC$_{50}$ = 0.3 nM), which combined several optimized residues and displayed binding affinities ($K_D$ = 0.4 nM) equivalent to that of the clinical Fab RG6147 ($K_D$ = 0.6 nM)[44]. Moreover, preliminary stability studies demonstrated that the potent inhibitors 1G10 and its derivative 1G10-ENI are highly stable (Supplementary Fig. 13).

Our findings underscore the importance of enriching the diversity of phage-displayed CKP libraries by including scaffolds with distinct native topologies and multiple loops that were available for protein interaction. Libraries of the carboxypeptidase A1 inhibitor scaffold yielded HTRA1 inhibitors which displayed an intriguing mechanism of inhibition featuring a cryptic binding pocket in the HTRA1 active site region. It is conceivable that the cryptic pocket discovered here could potentially be exploited for the discovery of small molecule inhibitors of HTRA1. The negative outcome of a recent clinical study testing the anti-HTRA1 Fab RG6147 for the treatment of geographic atrophy (Ph2 Gallego study)[48] challenges the view that HTRA1 activity is critical for disease etiology. Consistent with this notion are the conclusions of a recent study on HTRA1 expression in patient specimen, showing that age-related macular degeneration was not associated with increased levels of HTRA1[67]. Notwithstanding, the herein described potent and selective CKP inhibitors, which also inhibit mouse HTRA1, may serve as valuable tools to further investigate HTRA1 biology and could have other pharmacological applications in HTRA1-associated diseases, such as osteoarthritis and osteoporosis.

## Methods

### Recombinant HTRA proteases

The protease domains of human HTRA1 (HTRA1$^{PD}$; D161-K379) and mouse HTRA1 (mHTRA1$^{PD}$; D161-K379) contained an N-terminal His6 tag followed by a TEV protease cleavage sequence as described recently[47]. The chimeric HTRA1$^{PD}$-A202Y was made by introducing an A202Y mutation to the human HTRA1$^{PD}$ construct. The catalytically inactive S328A form HTRA1$^{PD(SA)}$ and HTRA1$^{PD/PDZ}$ (D161-P480) had an N-terminal His6 followed by a thrombin cleavage sequence[38]. The protease-PDZ domains of human HTRA3 (HTRA3$^{PD/PDZ}$; L130-M453) and HTRA4 (HTRA4$^{PD/PDZ}$; G153-N476) contained a C-terminal thrombin cleavage sequence followed by a His6 tag[44]. The chimeric HTRA4$^{PD/PDZ}$-Y200A was made by introducing a Y200A mutation to the HTRA4$^{PD/PDZ}$ construct. These constructs were expressed in *E. coli* and purified as described[38,44,47]. Human full-length HTRA1 (HTRA1$^{FL}$, Q23-P480) with an N-terminal His6 tag and TEV protease cleavage sequence[38] was expressed in *Trichoplusia ni* cells and purified as described[38,47]. The purity of the trimeric HTRA proteins after size exclusion chromatography was assessed by SDS-PAGE and LC/MS. The N- or C-terminal His6 tags of all HTRA proteins were not removed for the biochemical assays (enzyme assays, SPR, active site labeling). Human HTRA2$^{PD/PDZ}$ (A134-E458) was from R & D Systems (catalog #1458-HT-100) and contained a C-terminal His6. Human HTRA3$^{FL}$ (M1-M453, catalog #ab134450) and HTRA4$^{FL}$ (M1-N476, catalog #ab134444) containing a C-terminal His6 tag were from Abcam.

### Other proteins and reagents

Recombinant human Dickkopf-related protein 3 (DKK3) (catalog # 1118-DK), biglycan (catalog # 2667-CM) and decorin (catalog # 143-DE) were purchased from R & D Systems. The EnzCheck® assay kit with fluorescence-quenched casein substrate (catalog # E6638) was purchased from Thermo Fisher Scientific. The fluorescence-quenched synthetic substrate Mca-IRRVSYSFK(Dnp)K (H2-OPT) was purchased from Innovagen (catalog # SP-5076). The fluorescent activity-based probe (ABP) TAMRA-LV-ABP was described previously[44]. The TAMRA-DPMFKLV-ABP (TAMRA-7mer-ABP) is a derivative of the DPMFKLboroV inhibitor described by ref. 40, and was produced by changing the C-terminal boronic acid to a diphenyl phosphonate warhead and appending the TAMRA fluorophore at the N terminus. The non-fluorescent DPMFKLV-ABP (7mer-ABP)[47] was used for SPR experiments and as positive control for selectivity assays with HTRA1-4. The following human proteases were used for enzymatic assays: carboxypeptidase A1 (catalog # 2856-ZN) and its substrate Ac-Phe-Thiaphe-OH (catalog # STP-3621-PI) were from R & D Systems and Peptides International, respectively. Neutrophil elastase (catalog # 16-

14-051200), cathepsin G (catalog # 16-14-030107), and proteinase 3 (catalog # 16-14-161820) were purchased from Athens Research and Technology. Coagulation Factor XIa (catalog # HCXIA-0160), Factor Xa (catalog # HCXA-0060), plasmin (catalog # HCPM0140), and thrombin (catalog # HCT-0020) were from Haematologic Technologies. Plasma kallikrein (catalog # HPKa2650) was from Enzyme Research Laboratories, urokinase (catalog # CC4000) from Chemicon, trypsin-3 (catalog # 3714-SE-010) from R&D systems, and chymotrypsin-C (catalog # SRP6509) from Sigma. Neutrophil serine protease 4 (NSP4), hepsin, and hepatocyte growth factor (HGFA) were produced as described previously[68–70]. Chromogenic substrates S-2288 (catalog # S820852), S-2765 (catalog # S821413), S-2302 (catalog # S820340), S-2444 (catalog # S820357), S-2366 (catalog # S821090), S-2238 (catalog # S820324), and S-2586 (catalog # S820894) were purchased from Diapharma. The substrates M4765 (catalog # M4765) and DTNB (catalog # 22582) were purchased from Sigma, ADG217L (catalog # ADG217L) from Sekisui Diagnostics GmbH, and BML P303 (catalog # BML P303-0005) from Enzo Life Sciences. The NSP4 substrate PK421 and inhibitor PK401 have been described[71]. Protease inhibitors PMSF (catalog # 10837091001), Leupeptin (catalog # L5793), and PAI-1 (catalog # A8111) were purchased from Sigma.

### Enzyme assays

The fluorescence-quenched casein-BODIPY® assay (EnzCheck®, Thermo Fisher Scientific) was carried out in TNC buffer (50 mM Tris pH 8.0, 200 mM NaCl, 0.25% CHAPS) at 37 °C in 96-well black optical bottom plates (Nalge Nunc Int., Rochester, NY). Enzymes were incubated with CKPs or DMSO control for 15 min prior to addition of casein substrate. The reactant concentrations were: 30 nM HTRA1$^{FL}$, 30 nM HTRA1$^{PD/PDZ}$, 30 nM HTRA1$^{PD}$, 30 nM HTRA2$^{PD/PDZ}$, 30 nM HTRA3$^{FL}$, 100 nM HTRA3$^{PD/PDZ}$, 30 nM HTRA4$^{FL}$, 300 nM HTRA4$^{PD/PDZ}$, 2 μM CKP, 5 μg/ml casein substrate and 0.25% DMSO. For selectivity assays, 0.5 mM 7mer-ABP was used as positive control. The kinetics of substrate cleavage were measured at 37 °C on a SpectraMax M5 microplate reader (Molecular Devices) with excitation of 485 nm and emission of 530 nm for 30 min (HTRA1 and HTRA2) or 5 h (HTRA3 and HTRA4). The determined initial rates of cleavage were expressed as percent of control (DMSO control) after subtracting the baseline rates (without enzyme).

Assays with the fluorescence-quenched synthetic substrate Mca-IRRVSYSFK(Dnp)K (H2-OPT)[38,60] were carried out essentially as described[38]. CKP stock solutions in DMSO were diluted in TNC buffer (dilution series for IC$_{50}$ determinations, or final concentration of 2 μM for selectivity assay or as indicated for HTRA1/4 mutants) and incubated with enzymes in 96-well black optical bottom plates for 60 min at 37 °C. For selectivity experiments we used the 7mer-ABP as inhibitor, which was incubated with the proteases for 15 min at 37 °C. A 10 mM stock solution of H2-OPT in DMSO was diluted with water, prewarmed at 37 °C and then added to the CKP-enzyme mixtures. The final concentrations in the reaction mixtures were: 1 nM HTRA1$^{FL}$, 3 nM HTRA1$^{PD/PDZ}$, 1 nM HTRA1$^{PD}$, 1 nM HTRA2$^{PD/PDZ}$, 3 nM HTRA3$^{FL}$, 60 nM HTRA3$^{PD/PDZ}$, 6 nM HTRA4$^{FL}$, 1 μM HTRA4$^{PD/PDZ}$, 1 nM mouse HTRA1$^{PD}$, 25 nM HTRA1$^{PD}$-A202Y, 1 μM HTRA4$^{PD/PDZ}$-Y200A, 0.5 mM 7mer-ABP, 5 μM H2-OPT, and 0.25% DMSO. The kinetics of substrate cleavage was measured at 37 °C on a SpectraMax M5 microplate reader with excitation of 328 nm and emission of 393 nm. The determined initial rates of cleavage were expressed as percent of control (DMSO control) after subtracting the baseline rates (without enzyme). IC$_{50}$ values were determined by Sigmoidal 4PL modeling of multiple data points from serial dilutions of CKPs using Prism 9 software (GraphPad). For selectivity assays with HTRA1-4$^{PD/PDZ}$ constructs, the specific activities of the enzymes and enzyme:CKP inhibitor ratios were as follows: HTRA1$^{PD/PDZ}$, 12560 mRFU/min/nM – 1:667; HTRA2$^{PD/PDZ}$, 32300 mRFU/min/nM – 1:2000; HTRA3$^{PD/PDZ}$, 962 mRFU/min/nM – 1:33; HTRA4$^{PD/PDZ}$, 18 mRFU/min/nM – 1:2.

Assays with endogenous HTRA1 derived from C32 melanoma cells were performed essentially as described[46]. The C32 cell line was derived from the skin of a 53-year-old Caucasian male with amelanotic melanoma (ATCC #CRL-1585). C32 cells were cultured in DMEM (high glucose) supplemented with 10% FBS and 10 mM HEPES pH7.2 to 60% confluency then changed to the same medium without FBS for 3 days. The conditioned medium containing secreted HTRA1 was collected and supplemented with Roche cOmplete inhibitor cocktail (0.5 tablet to 15 ml medium), which inhibits the activity of generic proteases but not of HTRA1[46]. After removing cellular debris by centrifugation, the medium was then concentrated using a 10 kDa cut-off filter (Amicon) and stored in 10% glycerol at −80 °C. For activity assays the medium was tenfold diluted in TNC buffer and incubated with the CKPs (final concentration of 2 µM) or IgG94 (final concentration of 0.5 µM; positive control) for 60 min at 37 °C. IgG94 is anti-HtrA1 antibody developed in house and has been published before[46]. Prewarmed H2-OPT was added and activities determined as described for the H2-OPT assay.

Carboxypeptidase A1 assays were performed according to manufacturer's protocol. Briefly, carboxypeptidase A1 (100 µg/ml) was activated by incubation with 1.0 µg/ml trypsin (bovine trypsin catalog # T-1426, Sigma) in TNCB buffer (50 mM Tris, 0.15 M NaCl, 10 mM CaCl$_2$, 0.05% Brij-35, pH 7.5) at 25 °C for 60 min and 500-fold diluted in TNCB buffer. Then the CKPs were added and incubated for 15 min and equal volumes of TNCB buffer containing substrate (Ac-Phe-Thiaphe-OH) and DTNB reagent (5,5'-dithio-bis[2-nitrobenzoic acid]; catalog # T-1426, Sigma) were added. The final reactant concentrations in this mixture were: 0.1 µg/ml carboxypeptidase A1, 0.5 µM and 2.0 µM CKPs, 100 µM substrate and 100 µM DTNB. Substrate cleavage was immediately measured on a SpectraMax M5 for 5 min at 405 nm absorbance.

For experiments with the panel of 14 trypsin fold serine proteases, the CKP inhibitors 3B3 and 1A8 and control inhibitors were incubated with the proteases in assay buffer for 15 min at 37 °C. Prewarmed (37 °C) substrates were added and enzyme activity was immediately measured on a SpectraMax M5 microplate reader (Molecular Devices) at 405 nm (cleavage of the fluorescence-quenched NSP4 substrate PK421 was measured at 355 nm excitation and 460 nm emission with a cutoff of 455 nm). The determined initial rates of substrate cleavage were expressed as percent of control (DMSO control) after subtracting the baseline rates (without enzyme). The assay buffer for elastase and HGFA was 20 mM HEPES, pH 7.5, 150 mM NaCl, 5 mM CaCl$_2$, 1% BSA. The assay buffer for cathepsin G was 160 mM Tris 7.5 1.5 M NaCl, 0.5% BSA. The assay buffer for all other proteases was 50 mM Tris-HCl, 100 mM NaCl, pH 8, 0.1% Triton X-100. For experiments with cathepsin G and proteinase 3 the substrate BML P303 was first mixed with DNTB and then added to the proteases. The final concentrations of proteases and substrates in the reaction mixture were as follows: 3 nM hepsin− 0.5 mM S2366, 3 nM plasmin · 0.5 mM S2366, 20 nM HGFA · 0.5 mM ADG217L, 4 nM NSP4 · 0.05 mM PK421, 6 nM Factor XIa · 0.5 mM S2288, 10 nM urokinase · 0.3 mM S2444, 0.5 nM Factor Xa · 0.3 mM S2765, 0.25 nM plasma kallikrein · 0.6 mM S2302, 80 nM neutrophil elastase · 1 mM M4765, 20 nM cathepsin G · 0.5 mM BML P303 and 0.17 mM DNTB, 10 nM proteinase 3 · 0.5 mM BML P303 and 0.17 mM DNTB, 1 nM chymotrypsin · 0.3 mM S2586, 10 nM trypsin · 0.5 mM S2222, 1 nM thrombin · 0.05 mM S2238. The final concentration of CKPs was 2 µM. The final concentrations of control inhibitors were: 0.1 µM PK401 for NSP4, 0.4 mM Leupeptin for trypsin, hepsin, plasmin, HGFA, Factor Xa, and plasma kallikrein, 0.7 µM PAI-1 for Factor XIa and urokinase, and 1 mM PMSF for neutrophil elastase, cathepsin G, proteinase 3, chymotrypsin and thrombin. The specific activities of the enzymes were as follows: hepsin, 5.2 mOD/min/nM; plasmin; 5.5 mOD/min/nM; HGFA, 1.2 mOD/min/nM; NSP4, 1.9 mRFU/min/nM; F.XIa, 4.1 mOD/min/nM; urokinase, 1.4 mOD/min/nM; F.Xa, 49.2 mOD/min/nM; plasma kallikrein, 49.5 mOD/min/nM; neutrophil elastase, 0.7 mOD/min/nM; cathepsin G, 0.9 mOD/min/nM; proteinase 3, 2 mOD/min/nM;

chymotrypsin, 5.9 mOD/min/nM; trypsin, 0.8 mOD/min/nM; thrombin, 45 mOD/min/nM. The enzyme:substrate ratios ranged from 1:12.5 × 10$^3$ (NSP4, neutrophil elastase) to 1:2.4 × 10$^6$ (plasma kallikrein). For all enzymatic assays, the determined values represent the mean ± S.D. of at least three independent experiments.

## HTRA1 active site labeling with TAMRA-labeled activity-based probes

Labeling of HTRA1$^{PD}$ with fluorescently labeled (tetramethylrhodamine, TAMRA) activity-based probes (ABP) having a phosphonate warhead were carried out essentially as described[44,47]. HTRA1$^{PD}$ (2 µM) was incubated with or without 40 µM CKPs in TNC buffer for 1 h at room temperature. 20 µM of TAMRA-labeled 7mer-ABP (TAMRA-7mer-ABP)[47] or TAMRA-labeled LV-ABP (TAMRA-LV-ABP)[44] were added to the mixture and further incubated for 1 h at room temperature. Reduced samples were electrophoresed on Bolt 12% Bis-Tris gels (Thermo Fisher Scientific) in 1x MES SDS Running buffer (Thermo Fisher Scientific) at 200 V for 20 min. The TAMRA fluorescence was detected with Cy3 filter on an Amersham Typhoon Imager (GE Life Sciences). The gels were then stained in SimplyBlue SafeStain (Thermo Fisher Scientific) and imaged on a Bio-Rad Gel Doc EZ Imager.

## Macromolecular substrate cleavage assays

DKK3 or decorin (4 µM) were incubated with 1 µM of HTRA1$^{FL}$ or 1 µM of HTRA1$^{PD/PDZ}$ in TNC buffer for 16 h at 37 °C in the presence or absence of CKPs (5 µM). Biglycan (4 µM) was incubated with 0.5 µM of HTRA1$^{FL}$ or HTRA1$^{PD/PDZ}$ in TNC buffer for 4 h at 37 °C in the presence or absence of CKPs (2.5 µM). Reduced samples were electrophoresed on Bolt 12% Bis-Tris gels (Thermo Fisher Scientific) in 1x MES SDS Running buffer (Thermo Fisher Scientific) at 200 V for 25 min. Gels were stained in SimplyBlue SafeStain (Thermo Fisher Scientific) and imaged on a Bio-Rad Gel Doc EZ Imager.

## Surface plasmon resonance assays

Surface plasmon resonance (SPR) experiments were performed using a Biacore S200 instrument at 25 °C. His-tagged HTRA proteins were captured on a CM5 sensor chip using the His-capture kit (GE Healthcare). Protein concentrations used for immobilization were as follows; HTRA1$^{PD(SA)}$, HTRA1$^{PD}$ and HTRA1$^{PD}$-ABP 2 µg/ml with 60-s injections at 10 µl/min, HTRA2$^{PD/PDZ}$, HTRA3$^{PD/PDZ}$, and HTRA4$^{PD/PDZ}$ 4 µg/ml with 60-s injections at 10 µl/min. Approximate immobilization levels of HTRA proteins were as follows; HTRA1$^{PD(SA)}$ 700 RU, HTRA1$^{PD}$ 900 RU, HTRA1$^{PD}$-ABP 900 RU, HTRA2$^{PD/PDZ}$ 500 RU, HTRA3$^{PD/PDZ}$ 500 RU, HTRA4$^{PD/PDZ}$ 500 RU. All CKPs were prepared in HBS-EP buffer (0.01 M HEPES pH 7.4, 0.15 M NaCl, 3 mM EDTA, 0.005% v/v Surfactant P20). All CKP concentrations were injected for 120 s at a flow rate of 30 µl/min. A reference channel was subtracted from the HTRA-captured channel.

Binding kinetics of peptides to HTRA1$^{PD(SA)}$ were determined by flowing six serial three-fold dilutions of peptide over the HTRA1$^{PD(SA)}$ immobilized surface using a single cycle kinetics model. CKP maximum concentrations were as follows: 1A7, 1A8, 1C10, 2D5, and 3A7, 300 nM; 1G10-ENI and 1D3-ENI, 100 nM; 3B3, 3000 nM; and CPI, 10,000 nM. Data were analyzed using GE BIA evaluation software using a 1:1 Langmuir binding model. The $K_D$ values, calculated from the determined $k_{on}$ and $k_{off}$ values, were the mean ± S.D. of 3–4 independent experiments.

CKP binding to active-site blocked HTRA1 was assessed by preincubating HTRA1$^{PD}$ with 500-fold molar excess of 7mer-ABP (DPMFKLV-phosphonate) for 48 h at room temperature. The thus formed 7mer-ABP:HTRA1$^{PD}$ complex, as well as untreated HTRA1$^{PD}$ (control) were then captured on a CM5 sensor chip using the His-capture kit (GE Healthcare). CKPs were injected at single concentrations for 120 s at a flow rate of 30 µl/min. Peptide concentrations were as follows: 1C10 and 2D5, 300 nM; 3B3, 3 µM.

Binding analysis of the CKPs to HTRA1$^{PD}$, HTRA2$^{PD/PDZ}$, HTRA3$^{PD/PDZ}$, HTRA4$^{PD/PDZ}$ were determined by flowing single 1 µM injections of CKPs

over the immobilized HTRA surface using a CM5 sensor chip using the His-capture kit (GE Healthcare).

## CKP phage library design and construction

The EETI-II and CPI native scaffolds were selected to generate phage libraries with diversity in two surface loops. The EETI-II libraries, which were designed as previously described[7], consisted of randomized Loop1 or Loop5 and of simultaneously randomized Loop1 and 5 sequences. The CPI libraries were based on the wildtype CPI, but with the first two amino acids (QQ) deleted, which helped CPI expression in our initial validation and had no effect on the folding. The CPI libraries consisted of randomized Loop2 or Loop5 and of simultaneously randomized Loop2 and 5 sequences. While the CPI-Loop2 retained its native length (5 residues), the length of CPI-Loop5 varied from 6 to 10 residues. The libraries were constructed by fusing the CKPs to the N terminus of the M13 major coat protein p8 with a gD tag at the N terminus of the CKPs[7]. All phagemids were transformed into E. coli and grown at 30 °C overnight to display CKP-g8 fusion proteins. The propagated phage was purified and resuspended in 1 ml PBT buffer (PBS, 0.5% BSA and 0.1% Tween20). The quality of all libraries was examined by sequencing and the sequence diversity of all libraries was determined to be >10^10.

## CKP phage library sorting and affinity maturation

**Initial selections.** Libraries containing the same CKP scaffolds (EETI-II or CPI) were grouped together for the primary biopanning against biotinylated HTRA1^PD(SA) protein. HTRA1^PD(SA) was biotinylated according to manufacturer's instructions using the EZ-Link™ Sulfo-NHS-LC biotinylation kit (ThermoFisher Scientific). Four rounds of standard binding selections were performed in the buffer containing 20 mM Tris pH 8, 150 mM NaCl, 0.25% CHAPS. The first two rounds of selections were performed with the biotinylated HTRA1^PD(SA) immobilized on streptavidin-conjugated magnetic beads. The third and fourth rounds of selections were performed with biotinylated HTRA1^PD(SA) immobilized on neutravidin and streptavidin-coated plates, respectively. The details of biopanning process were the same as described previously[7]. Resulting phage clones were screened in phage ELISA against the biotinylated HTRA1^PD(SA). Clones with high signal-to-noise ratio were subject to Sanger sequencing. Sequences that were identified multiple times in the Sanger sequencing were selected for chemical synthesis and further biochemical characterization.

**Affinity maturation.** A total of seven phage libraries were constructed based on different parent sequences. Five libraries were constructed based on the 3B3 sequence: two libraries were randomized with five and eight C-terminal residues of 3B3, and two libraries were randomized at Loop1 and Loop2 of 3B3; the fifth 3B3-based library carried roughly 50% of the wildtype sequences and 50% of randomized amino acids of Loop5. The changed amino acids of the hits obtained from screening the Loop1, 2, and 5 randomization libraries were incorporated into 1A8 for peptide synthesis. In addition, two sets of libraries with randomized N- and C-terminal sequences at various lengths were constructed based on the 1A8-Y27L sequence. The first set of libraries had fixed peptide sequences starting with Asp3 and N-terminal sequence and length variations of up to eight residues (preceding Asp3). The second set of libraries had fixed peptide sequences up to residue Trp34 and C-terminal sequence and length variations of up to eight residues (after Trp34). All libraries were C-terminally linked to the minor g3 coat protein of M13 phage, and were used to select against the biotinylated HTRA1^PD(SA) with four rounds of standard binding selections. Clones were picked and analyzed as described for the initial selections.

**Phage titration ELISA.** Phage titration ELISA was performed with 384-well plates that were coated with 2 μg/ml of neutravidin-linked biotinylated HTRA1^PD(SA) and incubated with serially diluted phage in PBS buffer containing 0.5% BSA, 0.05% Tween 20 for 1 h at 4 °C. Plates were washed and incubated with anti-M13 antibody (1:10,000, HRP conjugated, Creative Diagnostics, cat# CAB-655M) at room temperature for 30 min. Plates were then washed and developed with TMB substrate at room temperature. After quenching with 1 M H_3PO_4, absorbance at 450 nm was recorded to calculate the EC_50 values.

## Peptide synthesis

Linear peptides were synthesized using solid-phase synthesis with standard 9-fluorenylmethoxycarbonyl protocols[72]. The unnatural amino acid building blocks Fmoc-CaMe-Asp(OtBu)-OH, Fmoc-3-chloro-L-tyrosine, Fmoc-4-chloro-L-phenylalanine, Fmoc-4-fluoro-L-phenylalanine-OH, Fmoc-3,4-difluoro-L-phenylalanine, Fmoc-L-4-methylphenylalanine, Fmoc-L-homophenylalanine and Fmoc-3,5-dichloro-L-tyrosine were obtained from Combi-Blocks while Fmoc-3,5-Difluoro-L-tyrosine was obtained from AA Blocks Inc. Linear peptides were then purified and folded in the folding buffer at room temperature for 24 h with shaking. The two folding buffers used were 0.1 M ammonium bicarbonate, pH 9.0, 2 mM reduced glutathione, 0.5 mM oxidized glutathione, 4% DMSO, or 0.1 M ammonium bicarbonate, pH 8.0, 1 mM reduced glutathione, 50% DMSO. Folded CKPs were purified with a C18 reversed-phase HPLC column. The correct molecular weights of peptides were confirmed by LC/MS[7]. All CPI-derived peptides had an amidated C terminus for better folding.

## Protein crystallization

Expression of the catalytically inactive HTRA1^PD(SA) was performed using E. coli Rosetta 2 DE3, which were induced with 0.2 mM IPTG for 24 h at 37 °C. Purification of HTRA1^PD(SA) was carried out using an Ni-IMAC column with all buffers containing 10% (v/v) glycerol similar as described[38]. In addition, the His-tag was removed by using thrombin and the protein was further purified using reverse Ni binding followed by size exclusion chromatography on a Superdex 200 column equilibrated with 0.15 M NaCl, 20 mM Tris pH 8.0, 10% (v/v) glycerol. Co-crystallization with the CKP inhibitors required methylation of HTRA1^PD(SA) as described by ref. 40. Commercially available crystallization screens produced several hits for the 3B3 and 3A7 CKP complexes tested, but after several days to weeks the crystals began to dissolve, which may have been related to the presence of glycerol. New complexes were produced by using 1–1.5 mg/ml of peptide (5 molar excess over HTRA1^PD(SA)) in 0.15 M NaCl, 20 mM Tris pH 8.0, 10% (v/v) glycerol and incubating for 24 h at 4 °C. The complex was then dialyzed against 0.15 M NaCl, 20 mM Tris pH 8.0 using 3 K MW Slide-A-Lyzer MINI dialysis device (Thermo) overnight at 4 °C. Optimized conditions for CKP complexes with 3B3 and 3A7 were 15% PEG 20000, 0.1 M HEPES pH 7.5 in sitting drops at 18 °C using vapor diffusion. Block-like crystals formed after 2–7 days. Crystals were preserved using 2 M LiSO_4 and then flash-frozen with liquid nitrogen.

For crystallization of the 1A8:HTRA1^PD(SA) complex we started with the same conditions used for the 3B3:HTRA1^PD(SA) complex (15% (w/v) PEG 20000, 0.1 M HEPES pH 7.5) and the crystals were further optimized with the condition containing 6% (w/v) PEG20000, 0.04 M HEPES pH 7.5, 15% (w/v) PEG 3350, 0.06 M citric acid pH 3.5. The initial condition for the 1G10:HTRA1^PD(SA) complex was derived from the PEG/Ion high-throughput screen and was optimized to 15% (w/v) PEG 3350, 4% Tacsimate pH 6.0 and 0.1 M Mg_2SO_4. Crystals of both 1A8 and 1G10 complexes were cryoprotected using 30% (w/v) PEG 3350 and then flash frozen with liquid nitrogen.

## X-ray data collection and structure determination

Data for the 3B3:HTRA1^PD(SA) complex were collected at Advanced Light Source (ALS) 5.0.2 and processed with iMosflm[73] in the space group

P2₁2₁2₁. The datasets were anisotropically scaled to 3.1 Å resolution using STARANISO. Molecular replacement was performed using Phaser[74] with the starting model HTRA1 (PDB: 3NZI)[40]. The model was refined with Phenix[75] and manual model building was performed using Coot[76]. Density for the 3B3 peptide was clearly seen in the unbiased mfo-Dfc maps. The initial model used for each of the 3B3 protomers was the potato carboxypeptidase A1 inhibitor from PDB: 4CPA[10]. Data for the 3A7: HTRA1$^{PD(SA)}$ complex was collected at ALS 5.0.1 and processed with XDS[77] in the space group P2₁2₁2₁. The data were anisotropically scaled to 2.9 Å resolution using STARANISO. The 3A7 complex was solved in a similar fashion as the 3B3 complex. Data for the 1A8: HTRA1$^{PD(SA)}$ complex were collected at Stanford Synchrotron Radiation Lightsource (SSRL) 12-1 and processed with iMosflm[73] in space group P2₁2₁2₁. Following anisotropic scaling to 3.0 Å using STARANISO the 1A8 complex was solved using the 3B3 model. The 1G10 complex data were collected at the Advanced Photon Source at Argone National Laboratory at 24ID-C and processed with XDS[77] in the space group P2₁2₁2₁. Data were anisotropically scaled to 3.2 Å using STARANISO and molecular replacement was performed using Phaser[74] with search model PDB: 3NZI. The final model was reached following iterative manual model building with Coot[76] and data refinement using Phenix[75]. Protein structures were visualized using Pymol version 2.5.2.

## Structural model of the 1G10 N-terminal tyrosine residue

The experimental crystal structure of the 1G10:HTRA1$^{PD(SA)}$ complex was loaded into MOE (Molecular Operating Environment (MOE), 2022.02 Chemical Computing Group ULC, 1010 Sherbrooke St. West, Suite #910, Montreal, QC, Canada, H3A 2R7, 2023) and all objects were fixed. Visual inspection of the structure indicated that the N atom of the terminal proline residues in 1G10 peptides chain X and chain U pointed toward the surface of the protein (as opposed to the solvent for other peptides). A tyrosine residue was added to the N-terminal proline residues and minimized so that all dihedral angels were at their lowest energy confirmation.

## CKP stability assay

The CKPs 1G10 and 1G10-ENI were incubated in PBS containing 0.15% DMSO at a concentration of 0.3 mM (-1 mg/ml) at 37 °C for 30 days. Samples from two independently performed experiments were frozen at −20 °C until further analysis. Thawed samples were diluted in H₂O and injected into LC-MS (Agilent). From the HPLC UV trace (at 220 nm) the CKP peak area was normalized to the DMSO peak area from the same run. The values obtained from multiple HPLC runs were averaged and shown as dot plot.

## Reporting summary

Further information on research design is available in the Nature Portfolio Reporting Summary linked to this article.

## Data availability

The data that support this study are available from the corresponding authors upon request. The X-ray structures of the complexes are 8SDM (3B3:HTRA1$^{PD(SA)}$), 8SDP (3A7:HTRA1$^{PD(SA)}$), 8SE7 (1A8:HTRA1$^{PD(SA)}$) and 8SE8 (1G10:HTRA1$^{PD(SA)}$). All reagents are available from the lead contact under a material transfer agreement with Genentech. Source data underlying Figs. 1, 2, 4, 6, Supplementary Figs. 2, 3, 6, 7, 8, 10, 12, 13 and Supplementary Table 1 are provided. Source data are provided as a Source Data file. Source data are provided with this paper.

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

## Acknowledgements

We would like to thank Bob Lazarus for his valuable comments on the manuscript. Synchrotron at the ALS is supported by the Director, Office of Science, Office of Basic Energy Sciences (BES), of the U.S. Department of Energy (DOE) under contracts DE-AC02-05CH11231. The Berkeley Center for Structural Biology is supported in part by the NIH, NIGMS, and the HHMI. The SSRL Structural Molecular Biology Program is supported by the DOE Office of Biological and Environmental Research, and by the NIH, NIGMS (including P41GM103393).

## Author contributions

Y.L., Y.W., W.L. and Y.Z. designed CKP libraries, carried out phage display, interpreted results and wrote the manuscript; M.U. performed X-ray crystallography and solved structures and wrote manuscript; Y.D. analyzed and interpreted crystal structures and wrote the manuscript; W.T. performed macromolecular substrate cleavage assays and activity-based probe labeling; B.T. performed SPR experiments; Y.W. performed HTRA protease expression and purification, mutagenesis and enzymatic assays; X.G. carried out stability studies and quality control of synthetic peptides and wrote manuscript; C.G. built structural models and interpreted results and wrote manuscript; J.F. designed CKPs containing unnatural amino acids, interpreted results and wrote manuscript; R.N.H. and D.K. designed and coordinated the overall study and wrote the manuscript.

## Competing interests

All authors were employees of Genentech Inc., a for-profit institution, at the time when the studies were performed.
