## [Peer Review File · Nature Communications]

Cystine-knot peptide inhibitors of HTRA1 bind to a cryptic pocket within the active site regionREVIEWER COMMENTS

Reviewer #1 (Remarks to the Author):

This work by Li et al describes the development of peptidic inhibitors of the serine protease HTRA1 using phage display, their structural analysis by X-ray crystallography, and the identification of a cryptic binding pocket on the protease. In brief, the group lead by Kirchhofer have cloned cystine-knot phage display libraries, performed phage display selections, isolated binders, improved them by iterative affinity maturation cycles, and characterized them functionally and structurally in depth including by X-ray crystallography.

This is an impressive piece of work that has most likely taken several years of work and much manpower, and culminated in the discovery of a surprising inhibition mechanism. The target HTRA1 is of interest for drug development as it is involved in several diseases (though a recent clinical study with a mAb was disappointing). The developed highly potent and selective inhibitors (sub nM) are of great interest as a research tools. And most important, the discovery of the cryptic site, being a pocket that allows selective inhibition, potentially offers an opportunity to develop selective small molecule inhibitors and potentially an oral drug. The quality of the work appears very good. The results are nicely presented. I think that this comprehensive study will interest a broader audience and is a good fit for Nature Communications. I recommend the following changes to improve the presentation of the article.

Minor recommendations:

1. The authors mention in the results data about an antibody that imposed a similar conformational change on HTRA1 as the peptidic inhibitors. I think this antibody is mentioned in the introduction but it is not described. I recommend considering to describe this already in the introduction.
2. Results about phage selection: the library size is not indicated in the results section and I recommend to add this important number. Also, I recommend to show the exact format (randomized amino acids, which amino acids, etc.) of the library (e.g. in Fig 1a).
3. Protein used for phage display selections: ... used the catalytically inactive HTRA protease domain ... This sounds not logic: why catalytically inactive? I guess the emphasis in this sentence should be on the catalytic domain (and not the entire protein), rather than on the inactivity.
4. A major achievement is the identification of the cryptic binding site that is obtained through a conformational change that is also seen when the Fab15H6.v4 binds to protease. The authors compare the two structures in a SI Figure but as this similarity is so central, I recommend to show this in a main figure.
5. Inhibition data: the authors mostly report IC50s but I strongly recommend to convert the data into Kis. At least the best inhibitors should be described with as Kis.

6. Page 3, ...applied both libraries ... This was confusing to read as only one library was mentioned in the sentences before. I guess the library based on the EETI-II is meant but it is not named "library".

7. The authors mention only at the end of the article the recent disappointing clinical studies with anti-HTRA1 antibodies (last paragraph of discussion). It may be better to have this in the introduction.

Reviewer #2 (Remarks to the Author):

The manuscript submitted by Li and colleagues focuses on finding effective and specific HtrA1 inhibitors. Since altered levels and/or activity of HtrA1 have been linked to several pathological events, including cancer, familial ischemic cerebral small-vessel disease, Alzheimer's disease and age-related macular degeneration, this topic is of potential interest to a wider audience. The main finding of this manuscript is the identification of a CKP (cysteine knot peptide)-based inhibitor that specifically blocks the proteolytic activity of HtrA1 but no other serine proteases. Using several biophysical and biochemical methods in combination with the crystallization of HtrA1-inhibitors complexes, the authors have revealed the binding site of this inhibitor and identified some mechanistic links of this interaction. However, there are concerns about the presentation and analysis of the data that diminish enthusiasm for the current manuscript.

The main issues are as follows:

1) HtrA1-4 proteins used in this study to show and compare the specificity of CKP inhibitors have different domain compositions, e.g., HtrA1FL is a full length protein containing IGF-BD, KM (Kazal motif), PD and PDZ domains, HtrA1PD includes only the PD domain, while HtrA2, HtrA3 and HtrA4 contain PD and PDZ domains. Since the N-terminal and PDZ domains affect activity, oligomerization state and/or substrate binding, this can significantly affect the results of such analysis. It is also unclear why 1) the HtrA proteins used for enzymatic analysis contain "foreign" sequences, such as the thrombin cleavage site and His6TAG, while they can be easily cut off, 2) these sequences are at different ends (for HtrA1FL, HtrA1PD and HtrA2PD/PDZ they are at the N-terminus, but for HtrA3PD/PDZ and HtrA4PD/PDZ at the C-terminus). Perhaps this issue is not very important for HtrA2 because it has the lowest sequence homology scores, a different domain organization, localization and function compared to HtrA1, but the fusion to 134AVPS motif at the N-end of HtrA2PD/PDZ might significantly affect its substrate binding affinity. Do the authors confirm that tagging at the N-terminus does not affect the activity of HtrA2 against tested substrates compared to HtrA2 without any tag? Why were HtrA3PD/PDZ and HtrA4PD/PDZ which are potentially more similar to HtrA1 constructed to have "foreign" sequences at the C-terminus linked to the PDZ domain? What is the rationale behind this strategy? This makes the specificity comparison less reliable.

2) The graphs showing proteolytic activity in % are not informative enough (Fig. 4a, Fig. S6d, Fig. S6e), because each enzyme was used at a different concentration and each has a different activity towards the tested substrate. Is it possible to show the appropriate enzyme activity units for each enzyme to get a reliable comparison? What is the enzyme/substrate/inhibitor ratio in each of these tests?

3) To compare the inhibitory effect of CPK peptides on the proteolytic activity of HtrA proteins, the

authors used the synthetic substrate H2-OPT. Is this peptide cleaved by all HtrA1-4 proteins with the same efficiency? Is there any paper showing that HtrA4 can cleave the H2-OPT peptide? Why are the concentrations of HtrA1-4 proteins different in enzymatic comparison assays? Perhaps it would make more sense to use a generic substrate for HtrA proteases, such as casein, for this purpose?

4) The gel from casein-BIODIPY digestion could be included in supplementary data. Why 3B3 has been selected for further studies? – 3A7 has a stronger inhibitory effect than 3B3 (Fig S2c).

5) Why different RU values were applied for HtrA proteins in the surface plasmon resonance assay? What were the concentrations of HtrA used in this assay?

6) The authors verified the inhibitory effect of selected CKPs on the proteolytic activity of HtrA1FL against three known physiological substrates. Unfortunately, we are unable to observe any specific degradation products, and those highlighted in Fig. 2a correspond rather to the auto-cleavage of HtrA1FL. Shorter incubation times should be applied. It is also unclear why instead of HtrA1PD/PDZ, HtrA1FL was used for this assay - it has relatively low proteolytic activity (zymogen forms). How do we explain that CKP inhibitors prevent auto-cleavage of HtrA1FL (Fig 2a)? It would be nice to see the results of this assay for HtrA1PD/PDZ and HtrA3PD/PDZ as well. Another common physiological substrate of HtrA1-4, XIAP, could potentially be used to confirm this hypothesis.

7) The manuscript contains only in vitro experiments based on isolated, recombinant proteins without demonstrating any in cellulo application. The effect of CKPs peptide on the cleavage of physiological HtrA1 substrates could be tested in mammalian cells with overexpression of HtrA1 or HtrA1-A202Y. What is the toxicity of these CKP peptides to mammalian cells? Can they potentially be used for some treatment? If not, digestion of endogenous proteins from cell lysates by HtrA1 in the presence of CKP peptides followed by western blotting analysis could be considered. This would significantly increase the value of the presented finding.

Reviewer #3 (Remarks to the Author):

In this manuscript, Li et al. report the discovery of novel cystine knot peptides (CKPs) as potent and selective inhibitors of the serine protease HTRA1. Dysfunction of HTRA1 has been implicated in numerous human pathologies, including age-related macular degeneration, and consequently HTRA1 has been explored as a potential drug target extensively. There is a clear need of molecules that target and modify HTRA1 activity, but leave other HTRA family proteases as well as other Trypsin-like serine proteases unaffected.

Li et al. present the identification of CKPs based on phage display screens initially, which were extensively characterized structurally and functionally, allowing for the rational improvement of these inhibitors in several rounds.

This study contains very careful design and experimentation, exploring the mechanism of binding and inhibition in great detail, showing convincingly how the engagement of a cryptic binding site in the S1' site of HTRA1 leads to the stabilization of a catalytically non-competent conformation of the active site. Meticulous biochemical and structural analyses are provided to confirm that the most potent inhibitors synthesized display high selectivity towards HTRA1 in addition to their remarkable potency.

The presented data comprise all relevant controls and many important and mechanistically interesting control experiments, e.g. exchanging key residues between HTRA1 and HTRA4 to investigate the

contribution of individual residues in the active site to enable potent inhibition of proteolytic activity. The manuscript is very well written, with a clear train of thought and convincing argumentation, the representation of data is clear and easy to follow.

The mechanistic findings of the binding and inhibition modes of HTRA1 are highly interesting and contribute significantly to our understanding and future potential of HTRA1 activity modulation. Overall, this is an excellent study that deserves publication in Nature Communications with minor edits.

Minor comments

Do the CKP inhibitors also inhibit HTRA1 after it has been allosterically stimulated, potentially forming higher-order oligomers beyond trimeric HTRA1?

Since HTRA1 is a conformation specific protease, it would be interesting to see whether proteolysis of intrinsically disordered substrates such as e.g. tau is inhibited as efficiently as the protein substrates tested?

How far is it possible that a small molecule could be developed that targets the cryptic S1' pocket based on the most potent inhibitor that gains potency by also binding a neighboring protomer at a rather distant site? Will that fact impede such efforts?

Are CKPs cell penetrant and could be used also as an experimental tool to investigate the substrate spectrum and physiological roles of the HTRA1 protease in cells? This aspect could be discussed a little further.

Typos

l. 35 stresses

l. 218 does this refer to Fig 3 h?

l. 462 one '' too much

REVIEWER COMMENTS

Reviewer #1 (Remarks to the Author):

This work by Li et al describes the development of peptidic inhibitors of the serine protease HTRA1 using phage display, their structural analysis by X-ray crystallography, and the identification of a cryptic binding pocket on the protease. In brief, the group lead by Kirchhofer have cloned cystine-knot phage display libraries, performed phage display selections, isolated binders, improved them by iterative affinity maturation cycles, and characterized them functionally and structurally in depth including by X-ray crystallography.

This is an impressive piece of work that has most likely taken several years of work and much manpower, and culminated in the discovery of a surprising inhibition mechanism. The target HTRA1 is of interest for drug development as it is involved in several diseases (though a recent clinical study with a mAb was disappointing). The developed highly potent and selective inhibitors (sub nM) are of great interest as a research tools. And most important, the discovery of the cryptic site, being a pocket that allows selective inhibition, potentially offers an opportunity to develop selective small molecule inhibitors and potentially an oral drug. The quality of the work appears very good. The results are nicely presented. I think that this comprehensive study will interest a broader audience and is a good fit for Nature Communications. I recommend the following changes to improve the presentation of the article.

Minor recommendations:

1. The authors mention in the results data about an antibody that imposed a similar conformational change on HTRA1 as the peptidic inhibitors. I think this antibody is mentioned in the introduction but it is not described. I recommend considering to describe this already in the introduction.

We have added a short description of this antibody to the introduction and combined it with point 7. Bottom of page 4: “This raises the possibility that CKPs may impair catalysis by trapping HTRA1 in the inactive conformation. Such a mechanism was recently described for the allosteric anti-HTRA1 Fab15H6.v4, which stabilized the non-competent active site conformation and completely inhibited catalysis[54]. This Fab was clinically tested for the treatment of geographic atrophy but, disappointingly, did not slow disease progression over a 72-week treatment period [55].”

2. Results about phage selection: the library size is not indicated in the results section and I recommend to add this important number. Also, I recommend to show the exact format (randomized amino acids, which amino acids, etc.) of the library (e.g. in Fig 1a).

We have added the library size to the results (p.6). “These combined libraries, having sequence diversity of more than 10^{10} , generated an extensive range of potential binding epitopes that increased the probability to discover inhibitors of HTRA1 enzyme function.” Also, we have modified Fig.1a (shown below) to indicate the randomized residues (X5 and X8 libraries).

		1	2	3	4	5	6	7	8	9	10	11	12	13	14	15	16	17	18	19	20	21	22	23	24	25	26	27	28	29	30	31	32	33	34	35	36	37	38	39	40
Peptide	IC50 ± S.D. (nM)	N terminus					L1					L2					L3					L4					L5					C terminus									
3B3 (parent)	53.1 ± 1.5	H	A	D	P	I	C	N	K	P	C	K	T	H	D	D	C	S	G	A	W	F	C	Q	A	C	Y	Y	A	T	W	S	C	G	P	Y	V	G	-	-	-
X5 library		H	A	D	P	I	C	N	K	P	C	K	T	H	D	D	C	S	G	A	W	F	C	Q	A	C	Y	Y	A	T	W	S	C	X	X	X	X	X	-	-	-
X8 library		H	A	D	P	I	C	N	K	P	C	K	T	H	D	D	C	S	G	A	W	F	C	Q	A	C	Y	Y	A	T	W	S	C	X	X	X	X	X	X	X	X
X8 group (X8 library):																																									
1C12	7.6 ± 0.1	H	A	D	P	I	C	N	K	P	C	K	T	H	D	D	C	S	G	A	W	F	C	Q	A	C	Y	Y	A	T	W	S	C	G	Y	Y	P	Y	W	F	R
1E4	27.8 ± 0.7	H	A	D	P	I	C	N	K	P	C	K	T	H	D	D	C	S	G	A	W	F	C	Q	T	C	Y	Y	A	T	W	S	C	G	L	G	Y	R	N	A	S
1A8	1.9 ± 0.2	H	A	D	P	I	C	N	K	P	C	K	T	H	D	D	C	S	G	A	W	F	C	Q	T	C	Y	Y	A	T	W	S	C	G	W	G	L	R	Q	I	D
2D5	1.6 ± 0.1	H	A	D	P	I	C	N	K	P	C	K	T	H	D	D	C	S	G	A	W	F	C	Q	T	C	Y	Y	A	T	W	S	C	G	W	P	P	R	H	R	D
X5 group (X5 library):																																									
2C10	17.5 ± 1.4	H	A	D	P	I	C	N	K	P	C	K	T	H	D	D	C	S	G	A	W	F	C	Q	A	C	Y	Y	A	T	W	S	C	G	P	G	Q	P	-	-	-
2G1	24.2 ± 1.6	H	A	D	P	I	C	N	K	P	C	K	T	H	D	D	C	S	G	A	W	F	C	Q	A	C	Y	Y	A	T	W	S	C	G	P	G	A	G	-	-	-
1B12	22.0 ± 0.7	H	A	D	P	I	C	N	K	P	C	K	T	H	D	D	C	S	G	A	W	F	C	Q	A	C	Y	Y	A	T	W	S	C	G	P	G	Q	I	-	-	-
1H8	27.9 ± 1.5	H	A	D	P	I	C	N	K	P	C	K	T	H	D	D	C	S	G	A	W	F	C	Q	A	C	Y	Y	A	T	W	S	C	G	P	G	R	V	-	-	-
2F11	12.6 ± 0.6	H	A	D	P	I	C	N	K	P	C	K	T	H	D	D	C	S	G	A	W	F	C	Q	A	C	Y	Y	A	T	W	S	C	G	P	G	Y	K	-	-	-
1C10	7.3 ± 0.4	H	A	D	P	I	C	N	K	P	C	K	T	H	D	D	C	S	G	A	W	F	C	Q	A	C	Y	Y	A	T	W	S	C	G	P	G	R	Y	-	-	-
1A7	5.8 ± 0.3	H	A	D	P	I	C	N	K	P	C	K	T	H	D	D	C	S	G	A	W	F	C	Q	A	C	Y	Y	A	T	W	S	C	G	P	G	W	R	-	-	-

3. Protein used for phage display selections: ... used the catalytically inactive HTRA protease domain ... This sounds not logic: why catalytically inactive? I guess the emphasis in this sentence should be on the catalytic domain (and not the entire protein), rather than on the inactivity.

Agreed. We have changed this sentence to emphasize the use of protease domain vs the full length HTRA1 (p.6). “Since the N- and the PDZ domains of HTRA1 do not contribute to enzyme activity [45, 47], we used the HTRA1 protease domain trimer (HTRA1^{PD(SA)}; S328A mutant) for library screening.”

4. A major achievement is the identification of the cryptic binding site that is obtained through a conformational change that is also seen when the Fab15H6.v4 binds to protease. The authors compare the two structures in a SI Figure but as this similarity is so central, I recommend to show this in a main figure.

We have moved the Fab15H6.v4 figure from the supplement section to the main figure section (it is now Fig. 3h). To make space for this figure, we have moved the previous Fig. 3g (showing the superpositions of competent with non-competent active site states) to the supplement section (it is now Suppl. Fig. S5d).

5. Inhibition data: the authors mostly report IC50s but I strongly recommend to convert the data into Kis. At least the best inhibitors should be described with as Kis.

We have never been able to generate good Michaelis-Menten curves with the H2-OPT substrate and therefore we were unable to determine a Km value which is required for converting the IC50s to Ki values. The substrate starts to precipitate at concentrations above 10 μM (at which the Vmax is not yet reached) and the velocity values decrease, which prevents Vmax and Km determinations. We explored various buffer systems and pegylated the substrate, but were unable to substantially improve the kinetics. That’s why we decided to present the data as the original IC50 values, which should be near the Ki values, since the relatively low substrate concentration are not expected to effectively compete with the CKP inhibitors. In support, the KD values determined by SPR are generally in good agreement with the IC50 values. We hope that this is acceptable to the reviewer.

6. Page 3,applied both libraries ... This was confusing to read as only one library was mentioned in the sentences before. I guess the library based on the EETI-II is meant but it is not named "library".

We made changes in the text (p.3) to improve clarity: "Recently, we employed a phage-displayed CKP library based on the *Ecballium elaterium* trypsin inhibitor II (EETI-II)[13] scaffold to identify LRP6 receptor binders that potently inhibited Wnt signaling [14, 15]. Here, we expanded the scaffold diversity by building a phage-displayed library based on the *potato carboxypeptidase A1* inhibitor (CPI) [16, 17]. We then applied both the EETI-II and CPI-based libraries for discovering inhibitors of the serine protease HTRA1 (High Temperature Requirement A1)."

7. The authors mention only at the end of the article the recent disappointing clinical studies with anti-HTRA1 antibodies (last paragraph of discussion). It may be better to have this in the introduction.

We are now mentioning the clinical study in the introduction (p.4/5), as suggested. "Such a mechanism was recently described for the allosteric anti-HTRA1 Fab15H6.v4, which stabilized the non-competent active site conformation and completely inhibited catalysis[54]. This Fab was clinically tested for the treatment of geographic atrophy but, disappointingly, did not slow disease progression over a 72-week treatment period [55]."

-Note: we have changed all bar graphs to show the individual data points as per Nat. Comm. policy.

Reviewer #2 (Remarks to the Author):

The manuscript submitted by Li and colleagues focuses on finding effective and specific HtrA1 inhibitors. Since altered levels and/or activity of HtrA1 have been linked to several pathological events, including cancer, familial ischemic cerebral small-vessel disease, Alzheimer's disease and age-related macular degeneration, this topic is of potential interest to a wider audience. The main finding of this manuscript is the identification of a CKP (cysteine knot peptide)-based inhibitor that specifically blocks the proteolytic activity of HtrA1 but no other serine proteases. Using several biophysical and biochemical methods in combination with the crystallization of HtrA1-inhibitors complexes, the authors have revealed the binding site of this inhibitor and identified some mechanistic links of this interaction. However, there are concerns about the presentation and analysis of the data that diminish enthusiasm for the current manuscript. The main issues are as follows:

1) HtrA1-4 proteins used in this study to show and compare the specificity of CKP inhibitors have different domain compositions, e.g., HtrA1FL is a full length protein containing IGF-BD, KM (Kazal motif), PD and PDZ domains, HtrA1PD includes only the PD domain, while HtrA2, HtrA3 and HtrA4 contain PD and PDZ domains. Since the N-terminal and PDZ domains affect activity, oligomerization state and/or substrate binding, this can significantly affect the results of such analysis. It is also unclear why 1) the HtrA proteins used for enzymatic analysis contain "foreign" sequences, such as the thrombin cleavage site and His6TAG, while they can be easily

cut off, 2) these sequences are at different ends (for HtrA1FL, HtrA1PD and HtrA2PD/PDZ they are at the N-terminus, but for HtrA3PD/PDZ and HtrA4PD/PDZ at the C-terminus). Perhaps this issue is not very important for HtrA2 because it has the lowest sequence homology scores, a different domain organization, localization and function compared to HtrA1, but the fusion to 134AVPS motif at the N-end of HtrA2PD/PDZ might significantly affect its substrate binding affinity. Do the authors confirm that tagging at the N-terminus does not affect the activity of HtrA2 against tested substrates compared to HtrA2 without any tag? Why were HtrA3PD/PDZ and HtrA4PD/PDZ which are potentially more similar to HtrA1 constructed to have “foreign” sequences at the C-terminus linked to the PDZ domain? What is the rationale behind this strategy? This makes the specificity comparison less reliable.

RE: use of constructs with N-domain and/or PDZ deletion: In order to alleviate concerns that the deletion of N-domain and PDZ domain affected the inhibitory activities of the CKPs or substrate interactions we are now presenting results for all HTRA proteases as full-length forms and as PD/PDZ forms for the H2-OPT substrate and for the macromolecular substrate casein (BODIPY assay). The casein substrate was used following the suggestion of this reviewer under point 3. Each assay tested the panel of 5 CKPs and the parent CPI scaffold (control). For the other macromolecular substrates (DKK3, Biglycan, Decorin) we now show in addition to HTRA1^{FL} the gels with HTRA1^{PD/PDZ} and HTRA2^{PD/PDZ} (see under point 6 below) The results from these different assays are very consistent and independent of the constructs used (full-length vs PD/PDZ); they demonstrate that the CKPs inhibited HTRA1 activity (both full-length and PD/PDZ forms), but not that of HTRA2-4 (both full-length and PD/PDZ forms). The results are presented in the new Suppl. Fig.S6a-c (and below). To align the presentation of the H2-OPT data with that of casein cleavage, we have modified the Fig.4a to show the H2-OPT results for the HTRA1-4^{PD/PDZ} constructs (shown below) (previously we showed full-length HTRA1 instead of HTRA1^{PD/PDZ}) and made a new Suppl. Fig. S6a to show the H2-OPT results for the HTRA1,3,4 full-length forms. We have updated the methods and results sections and figure legends accordingly.

NEW: Supplementary Figure S6

RE: tags at N- and C-terminus of HTRA proteases: The His-tags were not only important for protein purification (Ni NTA chromatography), but also for SPR binding studies in which the His-tag was used for immobilization on the chip surface. The decision to append the tags at the N- or C-terminus were mainly dictated by expression yields of the constructs, which sometimes differed significantly depending on the location of the tag. Even under the best conditions protein yields were small, except for HTRA1^{PD} and HTRA1^{PD(S/A)}. For some constructs we added a protease cleavage site that was specifically designed for potential structural applications allowing for tag removal for protein crystallizations. However, we only used the HTRA1^{PD(S/A)} for structure determinations in our current study and for this we removed the His6 tag. Regarding the HTRA2^{PD/PDZ} tag, we apologize for a mistake we made concerning the location of the His6-tag: this protein (from R and D Systems) actually had a C-terminal His6 tag and not an N-terminal His6-TEV cleavage site. This should alleviate concerns about fusion to the AVPS motif and we have corrected this in the methods section. The confusion arose because we also tested an N-

terminally tagged HTRA2^{PD/PDZ} construct in initial experiments (data not shown) the results of which were consistent with the C-terminally tagged construct; however, all experiments reported herein were done with the C-terminally tagged His6 construct.

We do not think that the N- or C-terminal location of the tags influences the activity of the proteases or the ability of the peptides to bind and inhibit. Based on our previous structural studies of HTRA1 (Eigenbrot 2012, ref#45), the N- and C-termini of all herein used forms (HTRA1^{PD}, HTRA1^{PD/PDZ}, HTRA1^{FL},) are too far away from the active site region to cause any disturbance in activity or peptide or substrate binding. The C-terminal tags of HTRA3^{PD/PDZ} and HTRA4^{PD/PDZ} are not expected to interfere with activity or peptide or substrate binding based on the structure of HTRA3^{PD/PDZ} (Glaza 2015, ref#56), which is also a suitable model for the not yet determined HTRA4^{PD/PDZ} structure. The HTRA3^{PD/PDZ} structure shows that the C-terminus of the PDZ domain containing a His6-tag is not in proximity to the active site and, therefore, is not expected to influence any potential peptide/substrate binding or activity. Glaza 2015 (ref#56), who solved the structure of HTRA3^{PD/PDZ} used both N- and C-terminal His6-tagged version of this protease in enzymatic assays and demonstrated excellent activities.

These conclusions were experimentally supported by the results of additional enzyme assays reported in this revised version, using full-length forms (all having C-terminal His6 tags) and PD/PDZ forms of HTRA1-4 which were tested in the H2-OPT substrate and the macromolecular substrate casein (see above). As a final note, our use of tagged HTRA proteases for enzymatic and binding studies is not unusual, but is in line with the general use of N- or C-terminally tagged recombinant HTRA proteases in the field. Numerous groups are commonly using N- or C-terminally tagged HTRA proteases for enzymatic assays without removing the tags (Trubestein et al., 2011, ref#47; Jones et al. PNAS 108:14578, 2011; Glaza et al., 2015, ref#56; Grau 2006, ref#35; Tennstaedt et al. JBC 287:20931, 2012; Martins et al., 2003, ref#71).

2) The graphs showing proteolytic activity in % are not informative enough (Fig. 4a, Fig. S6d, Fig. S6e), because each enzyme was used at a different concentration and each has a different activity towards the tested substrate. Is it possible to show the appropriate enzyme activity units for each enzyme to get a reliable comparison? What is the enzyme/substrate/inhibitor ratio in each of these tests?

We have now added the specific enzyme activities (mRFU/min/nM) and the enzyme:inhibitor ratios either to the methods section or to the figure legends (Fig.S6d - now it is Fig.S8d – and also to the new Supplementary Fig. S6a-c). We hope that this satisfies the reviewer's request for more transparency and clarity on the different enzyme activities towards the different substrates used. We have also added the enzyme:inhibitor ratios to the legend of Fig.S6e (now it is Fig.S8e), which shows the panel of 14 serine proteases. However, rather than showing each individual ratio in that figure, which would make for a cumbersome reading, we show the range of the ratios from lowest to highest. The specific enzyme activity for each of these 14 proteases and the range of enzyme:substrate ratios (high to low) is shown in the method section.

3) To compare the inhibitory effect of CPK peptides on the proteolytic activity of HtrA proteins, the authors used the synthetic substrate H2-OPT. Is this peptide cleaved by all HtrA1-4 proteins with the same efficiency? Is there any paper showing that HtrA4 can cleave the H2-OPT peptide? Why are the concentrations of HtrA1-4 proteins different in enzymatic comparison assays? Perhaps it would make more sense to use a generic substrate for HtrA proteases, such as casein, for this purpose?

The H2-OPT substrate was initially identified by Martins et al. (2003, ref#71) as a substrate for HTRA2. We were the first to use it for HTRA1 (Eigenbrot 2012; ref#45) and Glaza et al. (2015; ref#56) used a slightly modified version for HTRA3. H2-OPT has not been published as an HTRA4 substrate. Since H2-OPT worked well as a substrate for HTRA1-3 we hoped it may also be suitable for HTRA4. We were able to set up HTRA4 activity assays allowing us to measure the inhibition by the CKPs; however, it was a poor substrate for HTRA4 requiring relatively high enzyme concentrations. The activities of the HTRA proteases towards H2-OPT varied and that's why we used different enzyme concentrations in the assay. As mentioned under the previous point 2, we have now added the specific enzyme activities to the methods/figure legends to indicate the differences in H2-OPT cleavage among the HTRA proteases. Moreover, as suggested by the reviewer and as already elaborated under point 1, we have used the casein cleavage assay to measure inhibition of an expanded panel of HTRA proteases, which includes the full-length forms of HTRA1,3,4 and the PD/PDZ forms of HTRA1-4 (HTRA1-4^{PD/PDZ}) (Suppl. Fig.S6b,c). The results across the different assays and with different HTRA constructs are consistent and demonstrate excellent selectivity for HTRA1 for all 5 tested CKPs.

4) The gel from casein-BIODIPY digestion could be included in supplementary data. Why 3B3 has been selected for further studies? – 3A7 has a stronger inhibitory effect than 3B3 (Fig S2c). The initially performed enzymatic assay to identify inhibitory CKPs suggested that 3B3 and 3A7 had equal potencies (Suppl. Fig.S2c) and given the fact that 3A7 and 3B3 shared a similar sequence in Loop5 (Suppl. Fig. S2b), we selected only one of them, 3B3, for further affinity maturation. Only in subsequent studies did we realize that 3A7 had better binding affinity by SPR. At that time we had already performed the affinity maturation with our C-terminal extension libraries of 3B3 and obtained potent inhibitors (Fig. 1a) and there was no need to go back and affinity-mature 3A7.

We have tried to visualize the casein-BODIPY cleavage by gel fluorescence imaging (the amount of the casein-BODIPY substrate is too small to be seen as protein bands). The problem is that there are numerous bands ranging from >200kDa to <3kDa, which makes it very difficult to assess the inhibitory activities of 3B3 and 3A7. The manufacturer (Thermo Fisher Scientific) does not disclose any specifics about the substrate composition, which is likely a mixture of a casein(s) with different amounts of attached fluorophores. The imaged gel indicates that both 3B3 and 3A7 inhibited cleavage by HTRA1^{FL} and HTRA1^{PD}, albeit incompletely as expected from the enzyme assay results (Suppl. Fig. S2c). The reduced fluorescent band intensities of degradation products <6kDa in the 3B3 and 3A7 lanes are indicated in the gel below (asterisks). However, the gel is difficult to interpret due to the presence of numerous casein species and it will not be edifying to the reader. Moreover, we only pursued 3B3 and have much more clearcut and convincing gel photos from DKK3, biglycan and decorin digestions by HTRA1^{FL} and HTRA1^{PD/PDZ} (new Suppl. Fig. S3b), which not only show inhibition by 3B3 but by a panel 3B3-derived affinity-matured CKPs; therefore we chose not to show these data (below).

Protocol for figure above: The CKPs were incubated with HTRA1^{FL} or HTRA1^{PD} in TNC buffer (50 mM Tris pH 8.0, 200 mM NaCl, 0.25% CHAPS) for 15 min at 37°C and casein substrate was added. The reactant concentrations were: 30 nM HTRA1^{FL} or HTRA1^{PD}, 2 μM CKP and 5 μg/ml casein-BODIPY substrate. After 4 h at 37°C reduced SDS samples were electrophoresed on Bolt 12% Bis-Tris gels in 1x MES SDS Running buffer at 200 V. BODIPY Fluorescence was imaged on iBright1500.

5) Why different RU values were applied for HtrA proteins in the surface plasmon resonance assay? What were the concentrations of HtrA used in this assay?

For all binding kinetic analysis, the HTRA1 proteins (PD(SA), PD, and PD-ABP) were all immobilized at an identical concentration of 2 μg/mL with 60-second injections at 10 μL/min flow rate. The small differences in the immobilization levels between the PD(SA) and PD proteins may be attributed to slight differences in their His-capture. For the HTRA specificity binding studies, HTRA2/3/4^{PD/PDZ} proteins were immobilized at 4 μg/mL with 60-second injections at 10 μL/min flow rate. Further clarifications regarding the specific protein concentrations used have been included in the methods section for completeness and reproducibility (highlighted in red).

6) The authors verified the inhibitory effect of selected CPKs on the proteolytic activity of HtrA1FL against three known physiological substrates. Unfortunately, we are unable to observe any specific degradation products, and those highlighted in Fig. 2a correspond rather to the auto-cleavage of HtrA1FL. Shorter incubation times should be applied. It is also unclear why instead of HtrA1PD/PDZ, HtrA1FL was used for this assay - it has relatively low proteolytic activity (zymogen forms). How do we explain that CKP inhibitors prevent auto-cleavage of HtrA1FL (Fig 2a)? It would be nice to see the results of this assay for HtrA1PD/PDZ and HtrA3PD/PDZ as well. Another common physiological substrate of HtrA1-4, XIAP, could potentially be used to confirm this hypothesis.

-RE: visualization of cleavage products: Besides the HTRA1 autocleavage bands (40-50 kDa range) there are multiple degradation products visible (below 20kDa). These degradation bands are only seen in lane1 (the control w/o CKP) and in lane7 with the non-inhibitory CPI scaffold. The appearance of these degradation bands correlates with the complete disappearance of the intact substrate (filled arrows), which unambiguously demonstrates that the protein substrates have been fully degraded. In contrast, in the presence of the 5 inhibitory CKPs (lanes2-6) the substrate remained completely intact, which is clearly visualized in the gels (filled arrows). We have now increased the contrast on the gels to better visualize the degradation bands (indicated in red boxes).

Figure above: Indicated in red boxes are the degradation products, which are only present in the control lane (#1) and in the “CPI” lane (lane#7), but not in the presence of the 5 tested CKPs (lanes #2-6)

Below is an expanded gel pictures which shows additional low molecular degradation products as well as the CKP bands, which sometimes overlap with the degradation product bands. The gel will be uploaded as an xls file (“source data”) to *Nat. Comm.* and will be accessible.

-RE: HTRA1^{FL} having lower proteolytic activity than HTRA1^{PD/PDZ}: In our hands the HTRA1^{FL} is slightly more active than HTRA1^{PD/PDZ} in cleaving DKK3, Biglycan and Decorin (compare HTRA1^{FL} in Fig.2a with HTRA1^{PD/PDZ} in Suppl. Fig.S3b). In agreement, HTRA1^{FL} is 1.6-3 fold more active in the casein and H2-OPT assays. As requested here and under points 3, we have now added the results with HTRA1^{PD/PDZ} in the H2-OPT assays, in casein cleavage assays and in the DKK3/Biglycan/Decorin cleavage assays to the ms (Fig.4a, Suppl. Fig.S3b and S6b). The results show that there is no difference in the inhibitory activities of the 5 tested CKPs for HTRA1^{PD/PDZ} vs HTRA1^{FL}.

To our knowledge there are no zymogen forms in the HTRA family, including HTRA1. The autocatalytic cleavage of HTRA1 resulting in the removal of the Ndomain (IGFBP/Kazal) is not an activation event that is required for endowing HTRA1 with proteolytic activity as shown by several groups (Truebestein 2011, ref#47; Eigenbrot 2012, ref#45 ; Risor 2014, ref#63) and in the results presented herein. Therefore, the HTRA proteases differ from most other trypsin fold serine proteases which require an activation cleavage step that removes a propeptide.

-RE: CKPs prevent HTRA1 autocleavage (Fig.2a): HTRA1 is known to have autocatalytic activity (Risor 2014, ref#63) and we have observed this more than a decade ago when we noticed that the catalytically inactive HTRA1^{FL} (HTRA1^{FL(S/A)}) remained intact – visible as a single band by SDS-PAGE (~50 kDa), whereas several lower MW fragments (40-50 kDa range) were seen for the active wildtype form. Risor et al. 2014 (ref#63) identified the cleavage site to reside in the N-terminal IGFBP/Kazal domain (aka the N-domain), and autocatalysis can lead to the removal of the entire IGFBP/Kazal domain. This produces the HTRA1^{PD/PDZ}, which seems to represent a physiologically relevant form (Lorenzi 2013; ref#62). For our recombinant protein the autocatalysis appears to occur during purification, but not during storage at -20°C/-80°C; however, it is accelerated during the macromolecular cleavage reactions carried out at 37°C. By blocking the enzymatic (including autocatalytic) activity of HTRA1 in these experiments, the herein described CKPs prevent further HTRA1 autocatalytic cleavage as seen by the preservation of the intact 50kDa band in the Decorin and Biglycan gels (Fig.2a) (for DKK3 a major degradation product obscures this band).

-RE: HTRA1^{PD/PDZ} and HTRA3^{PD/PDZ} in macromolecular substrate assays. As suggested, we have carried out assays with HTRA1^{PD/PDZ} and HTRA3^{PD/PDZ} for all three macromolecular substrates and also for casein. We were unable to observe any appreciable cleavage of DKK3/biglycan/decorin by HTRA3^{PD/PDZ}, which however was able to cleave casein (results shown in new Suppl. Fig.S6b). The results of DKK3/biglycan/decorin by HTRA1^{PD/PDZ} are shown in the new Suppl. Fig.S3b and they are virtually identical to the results obtained with HTRA1^{FL} (Fig. 2a) in that all 5 CKPs inhibited the cleavage reactions, whereas the CPI scaffold – as expected – did not. The conditions (HTRA conc, CKP and substrate conc, time) were exactly the same as used for HTRA1^{FL} in Fig.2a. For DKK3 and biglycan, some of the CKP bands overlap with one of the low MW degradation bands. The results are now shown in the new Suppl. Fig. S3b and results, methods and figure legends were updated accordingly.

Instead of HTRA3^{PD/PDZ}, which did not cleave DKK3/biglycan/decorin we used HTRA2^{PD/PDZ} for experiments with these three macromolecular substrates in order to further assess CKP selectivity. There was relatively low activity towards DKK3 compared to the much more extensive cleavage activity for decorin and biglycan. Nevertheless, the results of 2 independent

experiments clearly demonstrated that the 5 CKPs did not inhibit cleavage of any of these three macromolecular substrates (representative gels shown below), which agrees with the casein cleavage results (Suppl. Fig. S6b,c) and supports the excellent selectivity of the CKPs for HTRA1. These results were not included in the ms.

Figure above: DKK3 or decorin (4 μ M) were incubated with 1 μ M of HTRA2^{PD/IDZ} in TNC buffer for 16 h at 37°C in the presence or absence of CKPs (5 μ M). Biglycan (4 μ M) was incubated with 0.5 μ M of HTRA2^{PD/IDZ} in TNC buffer for 4 h at 37°C in the presence or absence of CKPs (2.5 μ M). Reduced samples were electrophoresed on Bolt 12% Bis-Tris gels (Thermo Fisher Scientific) in 1x MES SDS Running buffer (Thermo Fisher Scientific) at 200 V for 25 min. Gels were stained in SimplyBlue SafeStain (Thermo Fisher Scientific) and imaged on a Bio-Rad Gel Doc EZ Imager.

Concluding remarks: We have now generated an extensive body of experimental evidence to firmly conclude that the CKPs are highly selective for HTRA1. The results from enzyme assays with different substrates are in complete agreement with SPR binding studies and with the molecular mechanism of inhibition identified by several high-quality crystal structures. The structures provide a rational explanation of the excellent selectivity, which is due to the non-conserved nature of the cryptic pocket within the HTRA family proteases and its absence in all other members of the trypsin fold serine proteases. These structural arguments were buttressed by experimental data using a large panel of serine proteases (Suppl. Fig. S8e) and by mutagenesis studies of cryptic pocket residues in HTRA1 and HTRA4 (Suppl. Fig. S8a-d).

7) The manuscript contains only in vitro experiments based on isolated, recombinant proteins without demonstrating any in cellulo application. The effect of CPKs peptide on the cleavage of physiological HtrA1 substrates could be tested in mammalian cells with overexpression of HtrA1 or HtrA1-A202Y. What is the toxicity of these CKP peptides to mammalian cells? Can they potentially be used for some treatment? If not, digestion of endogenous proteins from cell lysates by HtrA1 in the presence of CKP peptides followed by western blotting analysis could be considered. This would significantly increase the value of the presented finding.

RE: In cellulo application: HTRA1 having a signal sequence is a secreted protease and it is this extracellular form we wished to target in our efforts of generating CKP inhibitors. For example, it is the secreted HTRA1 form that is enriched in synovial fluid of osteoarthritis patients (Grau 2006, ref#35) and in the eye the secreted form of HTRA1 – the target for AMD therapy – is

found in the vitreous and the aqueous compartments (Tom 2020, ref#51). Therefore, our goal was develop peptide inhibitors that inhibit extracellular, rather than intracellular HTRA1 and these peptides were not expected to have cell permeability. This was borne out in experiments requested by another reviewer, demonstrating that the 3 selected CKPs (parent 3B3 and the two most potent CKPs 1D3-ENI and 1G10-ENI) had no detectable cell permeability in the MDCK cell assay.

RE: cell expressed HTRA1 vs recombinant HTRA1: To alleviate potential concerns regarding the use of recombinant HTRA1 proteins we have carried out experiments with endogenous cell-expressed HTRA1. For this we used the human melanoma cell line C32, which produces and secretes HTRA1 (full length form; Ciferri et al. 2015; ref#53). To reduce non-specific substrate cleavage the conditioned medium containing HTRA1 was supplemented with a protease inhibitor cocktail, which does not impair HTRA1 activity (Ciferri 2015; ref.53). The relatively low concentrations of HTRA1 in these preparations were only suitable for the H2-OPT assay, which is much more sensitive than the macromolecular substrate assays that require higher concentrations of HTRA1. As positive control we used the HTRA1-specific antibody IgG94 (Ciferri 2015; ref. 53). The results demonstrate that all CKPs, except the CPI scaffold, were able to inhibit enzyme activity of this endogenous HTRA1 form to a similar degree as the anti-HTRA1 antibody, suggesting that the CKPs recognize both endogenous and recombinant forms of HTRA1. The results are now shown in the new Suppl. Fig. S3a and results, methods and figure legends were updated accordingly.

Assays with endogenous HTRA1 derived from C32 melanoma cells were performed essentially as described (Ciferri 2015, ref#53). Conditioned medium (DMEM, 10 mM HEPES pH 7.2) containing secreted HTRA1 was collected after 3 days and supplemented with Roche cOmplete inhibitor cocktail (0.5 tablet to 15 ml medium). After removing cellular debris by centrifugation, the medium was then concentrated using a 10 kDa cut-off filter (Amicon) and stored in 10% glycerol at -80°C . For activity assays the medium was 10-fold diluted in TNC buffer and incubated with the CKPs (final concentration of $2\ \mu\text{M}$) or IgG94 (final concentration of $0.5\ \mu\text{M}$; positive control) (Ciferri 2015, ref#53) for 60 min at 37°C . Prewarmed H2-OPT was added and activities determined as described for the H2-OPT assay.

RE: toxicity of CKPs:

As requested, we have examined the cell toxicity of our CKPs in Hela cells and the C32 melanoma cell line. We found that the three CKPs 3B3, 1D3-ENI and 1G10-ENI even at very high concentrations ($50\ \mu\text{M}$) which were up to 100,000-fold above the K_D values, did not appreciably affect cell viability after 24h and 72h incubation, except for an up to 13% viability

reduction at the highest CKP concentration of 50 μ M after 72 h (see results below). The absence of cell toxicity is likely due to the inability of the CKPs to penetrate cell membranes, which was experimentally determined in response to another reviewer. Due to space limitations we have not included these results in the ms.

Protocol of cell toxicity assay (Figure above):

HeLa cells or C32 melanoma cells were passaged in DMEM media supplemented with 10% FBS, 1% pen-strep and 1x Glutamax using standard culture conditions (5% CO₂, 90% humidity at 37°C). At approximately 95% confluence cells were trypsinized, seeded onto 96-well plates (Corning) at 10,000 cells/well and grown for 24 h. Cells were then incubated with the CKPs 3B3, 1D3-ENI and 1G10-ENI (0, 0.6, 1.9, 5.6, 16.7 and 50 μ M) at 37°C for 24 h or for 72h. As a positive control we used staurosporine (STS; 1 μ M). Cell viability was measured using the CellTiter-Glo luminescent cell viability assay (Promega) on an Envision Multilabel plate reader. Average of at least three independent experiments \pm S.D. were plotted in Prism 10. Two-way ANOVA was performed on each dataset and compared to no drug samples and no significant viability reductions were observed, except for some treatments at the highest CKP concentration of 50 μ M: in HeLa cells at 24 h and at 72 h for 3B3 (9% and 13% reduced; p = 0.001 and 0.003) and in C32 cells at 72 h for 1D3-ENI and 1G10-ENI (13% reduction; p < 0.0001).

RE: potential use for treatment:

The good stability, selectivity and potency and the absence of apparent cell toxicity of the CKPs encourages the exploration of their therapeutic usefulness for HTRA1-associated diseases, such as osteoarthritis and perhaps osteoporosis. This is mentioned in the discussion section, last paragraph.

-Note: we have changed all bar graphs to show the individual data points as per Nat. Comm. policy.

Reviewer #3 (Remarks to the Author):

In this manuscript, Li et al. report the discovery of novel cystine knot peptides (CKPs) as potent and selective inhibitors of the serine protease HTRA1. Dysfunction of HTRA1 has been implicated in numerous human pathologies, including age-related macular degeneration, and consequently HTRA1 has been explored as a potential drug target extensively. There is a clear need of molecules that target and modify HTRA1 activity, but leave other HTRA family proteases as well as other Trypsin-like serine proteases unaffected.

Li et al. present the identification of CKPs based on phage display screens initially, which were extensively characterized structurally and functionally, allowing for the rational improvement of these inhibitors in several rounds.

This study contains very careful design and experimentation, exploring the mechanism of binding and inhibition in great detail, showing convincingly how the engagement of a cryptic binding site in the S1' site of HTRA1 leads to the stabilization of a catalytically non-competent conformation of the active site. Meticulous biochemical and structural analyses are provided to confirm that the most potent inhibitors synthesized display high selectivity towards HTRA1 in addition to their remarkable potency.

The presented data comprise all relevant controls and many important and mechanistically interesting control experiments, e.g. exchanging key residues between HTRA1 and HTRA4 to investigate the contribution of individual residues in the active site to enable potent inhibition of proteolytic activity. The manuscript is very well written, with a clear train of thought and convincing argumentation, the representation of data is clear and easy to follow.

The mechanistic findings of the binding and inhibition modes of HTRA1 are highly interesting and contribute significantly to our understanding and future potential of HTRA1 activity modulation. Overall, this is an excellent study that deserves publication in Nature Communications with minor edits.

Minor comments

1. Do the CKP inhibitors also inhibit HTRA1 after it has been allosterically stimulated, potentially forming higher-order oligomers beyond trimeric HTRA1?

We have carried out experiments to answer this question and closely followed the protocol of Truebestein et al. 2011 (ref#47), who were the first to publish the allosteric HTRA1 activation by denatured citrate synthase (CS). We were able to reproduce the allosteric activation, albeit to a lesser degree as published. The reason for this may be due to the different synthetic substrate (H2-OPT) we used. This substrate is very efficiently cleaved at low nM HTRA1 concentrations. Therefore, in order to measure proper enzyme kinetics (the HTRA1 concentration needs to be at or below 50nM for the initial reaction rates to be in the linear range) it necessitated a large dilution step of the initially formed HTRA1:CS complex (10 μ M:20 μ M). This dilution step may have led to the partial dissociation of the HTRA1-CS complexes, which resulted in a lower apparent HTRA1 activation. Nevertheless, we were able to reproducibly measure ~25% increased activation by CS and our CKP panel (the same as used in Figs. 2a and 4a) completely inhibited the activity of the preformed HTRA1:CS complex. We conclude that the CKPs are able to bind and fully inhibit the allosterically activated HTRA1 form. Due to space limitations (we are exceeding the allowed word count) and due to the addition of many new results to this

revised version, we decided not to include the CS results and hope this is agreeable to the reviewer.

+

Experiments were based on the Truebstein 2011 protocol: The complex of 10 μ M HTRA1^{PD} and 20 μ M citrate synthase (CS) was preformed in 100mM Tris-HCl pH 8.0, 150mM NaCl (assay buffer) and incubated for 10min at 42°C. The CKPs were added to a final concentration of 200 μ M (2% DMSO) and the reaction mixture was incubated for 30min at 42°C. Then the mixture was 200-fold diluted in assay buffer to give a final concentration of 50nM HTRA1^{PD} and 100nM CS. H2-OPT substrate was added (5 μ M final conc.) and the linear rates of substrate cleavage were measured at 42°C on a SpectraMax M5 microplate reader with excitation of 328 nm and emission of 393 nm and expressed as % of DMSO control (without CS).

2. Since HTRA1 is a conformation specific protease, it would be interesting to see whether proteolysis of intrinsically disordered substrates such as e.g. tau is inhibited as efficiently as the protein substrates tested?

Because we have no previous experience with Tau as an HTRA1 substrate, we chose the intrinsically disordered protein casein (Redwan et al. Disorder in milk proteins: caseins, intrinsically disordered colloids. Current Prot. Pept. Sci. 16: 228, 2015) as substrate, since cleavage can be readily quantified in the fluorescence-quenched casein-BODIPY® assay (EnzCheck®, Thermo Fisher Scientific), which we have used previously. We hope that this decision is agreeable to the reviewer. Since all HTRA proteases are capable of cleaving casein we also used this casein assay to address another reviewer's question regarding inhibitor selectivity. Therefore, the results of HTRA1 inhibition of the CKP panel (5 CKPs) towards the casein substrate are part of larger figures which also include HTRA2, HTRA3 and HTRA4 (both full length forms and PD/PDZ constructs (new Suppl. Fig.S6b and c). To align the presentation of the H2-OPT data with those of casein cleavage, we have modified the Fig.2a to show the H2-OPT results for the HTRA1-4^{PD/PDZ} constructs (previously we showed full-length HTRA1 instead of HTRA1^{PD/PDZ}) and made a new Suppl. Fig. S6a to show the H2-OPT results for the HTRA1,3,4 full-length forms. The results demonstrate that the panel of 5 CKPs, including the parent 3B3, inhibit the proteolytic activity of HTRA1^{PD/PDZ} as well as of HTRA1^{FL} towards casein (Suppl. FigS6b and c). The CPI scaffold was not inhibitory, as expected. As a positive control we used the activity-based probe (ABP), which inhibited all HTRA constructs. We conclude that the CKPs are able to inhibit HTRA1-mediated cleavage of the intrinsically disordered protein casein. The results of the two casein assays are shown below. We updated the methods, results and figure legends accordingly.

Supplementary Figure S6

3. How far is it possible that a small molecule could be developed that targets the cryptic S1' pocket based on the most potent inhibitor that gains potency by also binding a neighboring protomer at a rather distant site? Will that fact impede such efforts?

We have also thought about this possibility but we did not pursue any efforts with traditional small organic molecules. Instead, we carried out some pilot experiments with minimized peptides derived from the potent CKP 1A8, with the same idea, which is to interact with both, the cryptic pocket and the neighboring HTRA1 protomer. Since the 1A8-loop5 residues Y26, Y27 and W30 provide about 50% of the binding surface, we investigated the inhibitory activity of loop5-based cyclic peptides, which also contained the important C-terminal residue W34 residue that binds to the adjoining protomer. A small panel of designed peptides were synthesized and cyclized by formation of a thioether between the N-terminal chloro-acetylated residue and the C-terminal cysteine yielding peptides with ring sizes of 11-13 residues. We found that only two peptides inhibited HTRA1 enzyme activity with IC₅₀ values of 14 μM and 140 μM. It is likely that the constrained conformation of loop5 in the CKP 1A8 and the lack thereof in the cyclic peptide explains the poor potencies of the macrocycles. The more potent of these two peptides might be a starting point for engineering improved binding, but in the absence of any structural data we did not further pursue this approach. Based on these results and on structural

reasoning, we think that it will be extremely challenging to find chemical matter that is able to simultaneously engage with these two sites, other than with a highly constrained peptide. It may be more promising to use the cryptic S1' pocket as an anchor for a small molecule moiety and extend towards the proximal active site center, e.g. the S1 region. One might even consider an extension comprising a covalent reversible warhead, which would give a big boost in affinity, while the cryptic pocket binding may provide selectivity.

4. Are CKPs cell penetrant and could be used also as an experimental tool to investigate the substrate spectrum and physiological roles of the HTRA1 protease in cells? This aspect could be discussed a little further.

We carried out a cell permeability study using a MDCK trans-well assay, which is commonly used at Genentech for permeability screening of compounds. This assay measures the flux of test compound (Papp, apparent permeability) across a MDCK monolayer from the apical to the basolateral compartment over a 16 h period. For this we selected the parent 3B3 and the two most potent inhibitors 1D3-ENI and 1G10-ENI having pM binding affinities. Although we used very high CKP concentrations (up to 50 μM) (>100,000-fold above K_D for the 2 potent inhibitors), we were unable to detect any CKPs in the receiving (basolateral) chamber using a sensitive mass spectrometry method. The Papp values were $<0.003 \times 10^{-6}$ cm/s for the three CKPs ($n = 4-6$). For context, compounds with medium to high permeability have Papp values of $>1-10 \times 10^{-6}$ cm/s and compounds with Papp values of $<1 \times 10^{-6}$ cm/s are considered to be of low permeability. Therefore, we concluded that the 3 CKPs are cell impermeable and therefore they cannot be used to investigate the intracellular role of HTRA1.

The cell permeability results may explain the absence of cellular toxicity in studies, which we performed upon request of another reviewer. The results from that study demonstrated that these three CKPs 3B3, 1D3-ENI and 1G10-ENI even at very high concentrations of 50 μM, did not appreciably affect cell viability after 24h and 72h incubation, except for a slight viability reduction (up to 13%) at the highest concentration of 50 μM after 72 h. The most likely reason for the absence of toxicity is the inability of the CKPs to penetrate cell membranes. Due to space limitations and the lack of positive permeability results we have not added these data or elaborated on them in the ms.

Protocol of permeability assay:

Experiments were carried out as described (Chen et al. Evaluating the utility of canine Mdr1 knockout Madin-Darby canine kidney I cells in permeability screening and efflux substrate determination, *Mol Pharm.* 2018 Nov 5;15(11):5103-5113. doi: 10.1021/acs.molpharmaceut.8b00688. Epub 2018 Oct 10). Cells were seeded in 96-well Corning Transwell plates (PET membrane, 1 μm pore size) at 75,000 cells per well and grown in DMEM media supplemented with 10% FBS, 1% pen-strep and 1x Glutamax at 37°C, 5% CO₂ for 48 h to ensure monolayer formation. Stock solutions of the three CKPs 1B3, 1D3-ENI and 1G10-ENI were diluted in Hank's Balanced Salt solution containing 25 mM HEPES, pH 7.4 and 25 μM of the monolayer integrity marker lucifer yellow. CKPs at a concentration of 50 μM were added to the apical (donor) chamber and the CKP the apparent permeability (Papp) was determined by quantifying the amount of CKP in the basolateral (receiver) chamber after 16 h incubation at 37°C, 5% CO₂. The samples from the basolateral chamber were supplemented with the experiment analytical internal standard (100 nM propranolol in 50% ACN:50% water with 0.1% formic acid) and then injected in a Sciex Zeno TOF 7600 mass spectrometer using TOF/MS mode.

5. Typos

1. 35 stresses. Corrected.

l. 218 does this refer to Fig 3 h? yes, it refers to both Fig.3h (now Fig. 3g) and Fig. 3g (now Suppl. Fig. 4d). We have made the changes.

l. 462 one ‘.’ too much. Thank you for finding this typo; corrected.

-Note: we have changed all bar graphs to show the individual data points as per Nat. Comm. policy.

REVIEWERS' COMMENTS

Reviewer #1 (Remarks to the Author):

The authors have addresses all points criticized of which most were adjustments in presentation/text, with the exception of one related to IC50s where they convincingly explained why they do not indicate Kis (Km could not be determined due to solubility problems with substrate). As indicated before, this is a beautiful story and I can very much recommend to publish it!

Reviewer #2 (Remarks to the Author):

I find that the authors have put considerable effort to address most of the concerns. I am sure that these additional experiments and clarifications have markedly improved the manuscript and increased its merit and will be very useful to other groups working with human HtrA serine proteases. The use of proteins with the same domain organization and composition is crucial for a reliable comparison, and even such very simple experiments using a cell model as performed by the authors suggest the potential for application of the developed inhibitors. Based on my criticisms and suggestions they have made several changes in the text and figures and now provide a thoroughly revised and improved version of the manuscript.

The only minor points that could be additionally included are as follows:

- since HtrA3PD/PDZ has previously been shown to degrade decorin and biglycan (PMID 15206957), the results obtained in Rebuttal letter are somewhat surprising. Therefore, it could be worth to consider adding these results to the material available to readers in Supplementary or RAW data, even if the authors claim that they did not observe a positive degradation effect. HtrA2 is an intracellular protease, and decorin and biglycan are not considered HtrA2 substrates.
- that would be helpful to see the quality all of recombinant HtrA1-4 proteins, especially since some of them are no longer available (ab134450 and ab134444). Therefore, simple s SDS-PAGE of all HtrA proteins (FL, PD/PDZ) used in this study would be highly appreciated.
- cleavage of H2Opt by HtrA4 PD/PDZ was previously shown in PMID 31470122.

These minor suggestions do not affect the high value of the manuscript. I have no further comments and thus can recommend this work for publication in Nature Communications.

Reviewer #3 (Remarks to the Author):

The authors have addressed all concerns and minor comments very accurately and with significant effort and understanding. It was very pleasant to see the comments addressed in such a detailed and extremely informative way. As a result, there are no reservations regarding the publication of the revised version of the manuscript in Nature Communications.

Point-by-point response

Rev#3: I find that the authors have put considerable effort to address most of the concerns. I am sure that these additional experiments and clarifications have markedly improved the manuscript and increased its merit and will be very useful to other groups working with human HtrA serine proteases. The use of proteins with the same domain organization and composition is crucial for a reliable comparison, and even such very simple experiments using a cell model as performed by the authors suggest the potential for application of the developed inhibitors. Based on my criticisms and suggestions they have made several changes in the text and figures and now provide a thoroughly revised and improved version of the manuscript. The only minor points that could be additionally included are as follows:

- since HtrA3PD/PDZ has previously been shown to degrade decorin and biglycan (PMID 15206957), the results obtained in Rebuttal letter are somewhat surprising. Therefore, it could be worth to consider adding these results to the material available to readers in Supplementary or RAW data, even if the authors claim that they did not observe a positive degradation effect. HtrA2 is an intracellular protease, and decorin and biglycan are not considered HtrA2 substrates.

According to the rev#2 statement that these are “only minor points that could be additionally included” and “these minor suggestions do not affect the high value of the manuscript” we chose to include only one of the three points in the ms and hope that this is acceptable to the editor.

We observed that our HTRA3PD/PDZ protein has less than 10% of the activity of HTRA3FL, which may explain why we did not find appreciable cleavage of decorin/biglycan. We intended to repeat these experiments with the more active, commercially obtained HTRA3FL, but this product was discontinued and we ran out material. Therefore, we do not feel confident to challenge the previously published positive decorin/biglycan cleavage results (Tocharus 2004, ref#50) with our limited data set. By using the HTRA3PD/PDZ -cleavable macromolecular substrate casein instead of decorin/biglycan we were able to generate convincing inhibitor selectivity results presented in the recently submitted revised version, which satisfied the criticism raised by rev.#2.

- that would be helpful to see the quality all of recombinant HtrA1-4 proteins, especially since some of them are no longer available (ab134450 and ab134444). Therefore, simple s SDS-PAGE of all HtrA proteins (FL, PD/PDZ) used in this study would be highly appreciated. Most of the rec. HTRA proteins used in this study are already shown on gels in different figures (Fig.2a, Fig. S3b, Fig.S8c). We no longer have material of the discontinued HTRA3/4FL proteins. Therefore, there is no need for an extra SDS-PAGE gel figure of rec. proteins.

- cleavage of H2Opt by HtrA4 PD/PDZ was previously shown in PMID 31470122. We have now included the literature reference (Wenta et al. 2019; ref#68) concerning cleavage of the H2- OPT substrate by HTRA4PD/PDZ. We added it to the results section on page 10.

These minor suggestions do not affect the high value of the manuscript. I have no further comments and thus can recommend this work for publication in Nature Communications.